# Attention boosted Individualized Regression

**Guang Yang**
Department of Data Science
College of Computing
City University of Hong Kong
guang.yang@my.cityu.edu.hk

**Yuan Cao**
Department of Statistics and Actuarial Science
School of Computing and Data Science
The University of Hong Kong
yuancao@hku.hk

**Long Feng**[*]
Department of Statistics and Actuarial Science
School of Computing and Data Science
The University of Hong Kong
lfeng@hku.hk

## Abstract

Different from classical one-model-fits-all strategy, individualized models allow parameters to vary across samples and are gaining popularity in various fields, particularly in personalized medicine. Motivated by medical imaging analysis, this paper introduces a novel individualized modeling framework for matrix-valued data that does not require additional information on sample similarity for the individualized coefficients. Under our framework, the model individualization stems from an optimal internal relation map within the samples themselves. We refer to the proposed method as Attention boosted Individualized Regression, due to its close connections with the self-attention mechanism. Therefore, our approach provides a new interpretation for attention from the perspective of individualized modeling. Comprehensive numerical experiments and real brain MRI analysis using an ADNI dataset demonstrated the superior performance of our model.

## 1 Introduction

Model-based machine learning methods have advanced significantly and become essential in modern data analysis. From linear regression to deep neural networks, most approaches fundamentally follow an one-model-fits-all strategy, meaning that parameters of a well-trained model are fixed and do not change for different samples. However, in fields like medical diagnosis and treatment design, it is important to explore and apply individualized models with parameters tailored to each sample, adapting to their unique features. Due to the heterogeneity among instances, individualized models are expected to provide more accurate predictions and personalized interpretations, which are their main advantages.

Individualized modeling has been extensively investigated in research, with the earliest example possibly being the varying coefficient models [10, 6] in statistics community. A varying coefficient model usually includes an additional variable and represents the varying coefficient as a function of this extra variable. These models have been applied and adapted in various contexts. For instance, [8] explored spatial modeling using a spatially varying coefficient process. In a similar vein, [26] considered varying coefficient models in image response regression and proposed using deep neural networks to estimate the varying coefficients.

---

[*]Long Feng is the corresponding author.

Beyond varying coefficient models, recent studies have also incorporated prior knowledge of sample similarity to regulate sample-specific coefficients. The fundamental assumption is that the similarity among coefficients for different samples relies on the sample similarity, meaning that the more similar the samples, the closer their coefficients. For instance, [24] tackled personalized medical models using a multi-task learning approach called FORMULA, assuming that models for similar patients are close and achieving this through Laplacian regularization. [25] developed the localized Lasso, which assumes a known weighted network over samples that reflects the distance in parameter space. [15] loosened the aforementioned prior assumption by considering additional covariates and assuming the existence of some measurement of similarity corresponding to similarity in parameter space. Moreover, they constrained the matrix of personalized parameters to be low-rank, so closeness in loadings implies closeness in parameters. While effective in various contexts, these methods heavily rely on the prior knowledge about parameter similarity, which might not be readily available in numerous real-world applications.

This paper aims to develop a novel individualized modeling framework for matrix-valued data, without the need for additional information on sample similarities. In our framework, model individualization is derived from the heterogeneity inherent in the samples themselves. Specifically, we seek to find an optimal sample-specific internal relation map to enhance model fitting and interpretation. The sample-specific relation map allows us to capture the local dependence between patches within each matrix input, thereby enhancing prediction performance and model interpretability.

It is worth noting that the proposed individualized modeling framework with sample-specific internal relation map is highly connected to the self-attention mechanism [21], which has demonstrated its exceptional performance in various field, including natural language processing, computer vision, and more. Due to such connection, we named the proposed framework Attention boosted Individualized Regression (AIR). Therefore, our approach could also provide a new interpretation for attention from the perspective of individualized modeling.

We should emphasize that our proposed approach is particularly well-suited for applications in personalized medicine and brain connectomics analysis, which initially motivated us to study individualized modeling. In recent years, the field of brain connectomics has experienced rapid growth due to advancements in medical imaging technology. This area of study focuses on examining comprehensive maps of connections within the human brain, playing a vital role in cognitive neuroscience, clinical diagnosis, and more. Brain networks can be represented by relation matrices, often established based on connections among regions of interest (ROIs). Besides sample features, the internal relationships within each sample may also influence relevant responses. This consideration has been addressed in the literature, such as [18, 9]. In this context, differentiated internal relations can emphasize heterogeneity among subjects, providing individual-level information about brain connectivity. This effect supports the use of internal relations in individualized models and further contributes to personalized medicine.

## 2 Attention boosted individualized regression

Given any matrix $\boldsymbol{M}$, we first introduce a matrix reshaping operator that allows us to explore the internal relations within $\boldsymbol{M}$. Let $\boldsymbol{M}$ have dimensions $D_1 \times D_2$ and let $d_1, d_2$ be factors of $D_1, D_2$. Define $(p_1, p_2) = (D_1/d_1, D_2/d_2)$. We can now define the operator $\mathcal{R}_{(d_1,d_2)}(\cdot) : \mathbb{R}^{D_1 \times D_2} \to \mathbb{R}^{(p_1 p_2) \times (d_1 d_2)}$ as a mapping from $\boldsymbol{M}$ to

$$\mathcal{R}_{(d_1,d_2)}(\boldsymbol{M}) = \left[ \text{vec}\left( \boldsymbol{M}_{1,1}^{d_1,d_2} \right), \ldots, \text{vec}\left( \boldsymbol{M}_{p_1,p_2}^{d_1,d_2} \right) \right]^\top, \tag{1}$$

where $\boldsymbol{M}_{j,k}^{d_1,d_2}$ represents the $(j,k)$-th block of $\boldsymbol{M}$ with size $d_1 \times d_2$. The operator $\mathcal{R}_{(d_1,d_2)}(\cdot)$ essentially vectorizes each of the $p_1 \times p_2$ block (of size $d_1 \times d_2$) and stacks the vectorized blocks together. Denote the inverse operation of $\mathcal{R}_{(d_1,d_2)}(\cdot)$ as $\mathcal{R}_{(d_1,d_2)}^{-1}(\cdot)$. A special case occurs when $(d_1, d_2) = (1, D_2)$, in which case we have $\mathcal{R}_{(1,D_2)}(\boldsymbol{M}) = \mathcal{R}_{(1,D_2)}^{-1}(\boldsymbol{M}) = \boldsymbol{M}$. It is worth noting that this reshaping operation has also been applied in attention mechanisms, allowing us to examine the relations or correlations among the $p_1 \times p_2$ patches.

Suppose we observe $n$ samples with scalar outcomes $y_i \in \mathbb{R}$ and matrix inputs $\boldsymbol{X}_i^{\text{ori}} \in \mathbb{R}^{D_1 \times D_2}$ for $i = 1, \ldots, n$. Given a block size $(d_1, d_2)$, we first reshape the original images to obtain

$\boldsymbol{X}_i = \mathcal{R}_{(d_1,d_2)}(\boldsymbol{X}_i^{\text{ori}}) \in \mathbb{R}^{p \times d}$, where $p = p_1 p_2$ and $d = d_1 d_2$. Then we consider the following individualized linear regression model with coefficient matrices varying across samples

$$y_i = \langle \boldsymbol{X}_i, \boldsymbol{C}_i \rangle + \varepsilon_i, \ \ i = 1, \dots, n, \tag{2}$$

where $\boldsymbol{C}_i \in \mathbb{R}^{p \times d}$ is the unknown individualized coefficient matrix for $i$-th sample and $\varepsilon_i$ is the noise term. Note that the reshaping operation $\mathcal{R}_{(d_1,d_2)}(\cdot)$ is one-to-one. Thus, model (2) is equivalent to $y_i = \langle \boldsymbol{X}_i^{\text{(ori)}}, \boldsymbol{C}_i^{\text{(ori)}} \rangle + \varepsilon_i$, with $\boldsymbol{C}_i^{\text{(ori)}} = \mathcal{R}_{(d_1,d_2)}^{-1}(\boldsymbol{C}_i)$. As previously mentioned, model (2) type of individualized regression has been studied under various constraints on the individualized coefficients, such as [10, 6, 24, 25].

In this paper, we propose to model $\boldsymbol{C}_i$ with two components: a homogeneous coefficient $\boldsymbol{C}$ reflecting common effects and a heterogeneous coefficient $\boldsymbol{D}_i$ containing individualized information. Specifically,

$$\boldsymbol{C}_i = \boldsymbol{C} + \boldsymbol{D}_i, \ \ i = 1, \dots, n. \tag{3}$$

For the heterogeneous coefficients, we further assume that they share an unknown common factor $\boldsymbol{D}$ across samples,

$$\boldsymbol{D}_i = \boldsymbol{A}_i^\top \boldsymbol{D}. \tag{4}$$

Here, $\boldsymbol{A}_i$ represents unknown sample-specific factors serving as a re-weighting matrix to aggregate the coefficients in $\boldsymbol{D}$, where the transpose is to better connect with self-attention mechanism later. The matrix $\boldsymbol{D} \in \mathbb{R}^{p \times d}$ can be viewed as a base coefficient matrix for the heterogeneous effects. Clearly, additional constraints on the individual factor $\boldsymbol{A}_i$ are necessary to ensure the identifiability of the model. The choice of factor $\boldsymbol{A}_i$ may vary depending on the purpose. In this paper, we propose an internal-relation-boosted individualized factor for matrix-valued inputs. Specifically, we consider $\boldsymbol{A}_i \in \mathbb{R}^{p \times p}$ of the form

$$\boldsymbol{A}_i = g(\boldsymbol{X}_i \boldsymbol{W} \boldsymbol{X}_i^\top), \tag{5}$$

where $\boldsymbol{W} \in \mathbb{R}^{d \times d}$ is an unknown matrix to be learned, while $g(\cdot) : \mathbb{R}^{p \times p} \to \mathbb{R}^{p \times p}$ is a known function that preserves dimension, of which different forms to be discussed. It is worth recalling that $\boldsymbol{X}_i = \mathcal{R}_{(d_1,d_2)}(\boldsymbol{X}_i^{\text{ori}}) \in \mathbb{R}^{p \times d}$ is the reshaped matrix. The reshaping operation (1) enables us to calculate the "generalized correlation" between patches through (5). When fixing $\boldsymbol{W} = \boldsymbol{I}_d$ and setting $g(\cdot)$ as the identity function, $\boldsymbol{A}_i = \boldsymbol{X}_i \boldsymbol{X}_i^\top$ reduces to standard covariance matrix of patches within $i$-th sample when $\boldsymbol{X}_i$ is properly centered.

In the formulation (5), the individualized matrix $\boldsymbol{A}_i$ is designed to capture the internal relationships among the $p$ rows of reshaped matrix (or $p$ patches of original matrix) for each sample. Relations between two vectors can be measured in different ways, such as correlation, similarity, distance, etc. Our formulation of (5) is motivated by the rotation correlation introduced by [20]. For any two vectors $\boldsymbol{u}$ and $\boldsymbol{v}$, the rotation correlation is defined as

$$\max_{\boldsymbol{H}} \boldsymbol{u}^\top \boldsymbol{H} \boldsymbol{v},$$

where the matrix $\boldsymbol{H}$ is usually required to be orthogonal. This rotational correlation aims to find the maximized correlation between $\boldsymbol{u}$ and $\boldsymbol{v}$ with the best possible rotation. When $\boldsymbol{H}$ is the identity matrix and $\|\boldsymbol{u}\|_2 = \|\boldsymbol{v}\|_2 = 1$, the rotation correlation reduces to standard Pearson correlation. We note that the $(j, k)$-th element of the sample-specific factor can be written as $\{\boldsymbol{A}_i\}_{jk} = \{\boldsymbol{X}_i\}_{j.} \boldsymbol{W} \{\boldsymbol{X}_i\}_{k.}^\top$, where $\{\boldsymbol{X}_i\}_{j.}$ and $\{\boldsymbol{X}_i\}_{k.}$ are the $j$-th and $k$-th rows of $\boldsymbol{X}_i$, respectively. In other words, $\{\boldsymbol{A}_i\}_{jk}$ is related to the rotation correlation between $\{\boldsymbol{X}_i\}_{j.}$ and $\{\boldsymbol{X}_i\}_{k.}$. However, our goal is not to maximize the correlation between $\{\boldsymbol{X}_i\}_{j.}$ and $\{\boldsymbol{X}_i\}_{k.}$ but to find the optimal rotation that achieves the best fitting for the responses.

Combining (2) to (5), we obtain our individualized model in the following form

$$y_i = \underbrace{\langle \boldsymbol{X}_i, \boldsymbol{C} \rangle}_{\text{homogeneous}} + \underbrace{\langle \boldsymbol{X}_i, g(\boldsymbol{X}_i \boldsymbol{W} \boldsymbol{X}_i^\top)^\top \boldsymbol{D} \rangle}_{\text{heterogeneous}} + \varepsilon_i. \tag{6}$$

Here, $\boldsymbol{C} \in \mathbb{R}^{p \times d}$, $\boldsymbol{D} \in \mathbb{R}^{p \times d}$, and $\boldsymbol{W} \in \mathbb{R}^{d \times d}$ are the coefficient matrices that need to be learned. The decomposition of (6) allows us to understand and assess the individuation degree of each sample

and the entire model. At the sample level, a larger magnitude of the heterogeneous part indicates that the sample is more distinctive, affected by its internal relations. At the model level, the larger the magnitude of the homogeneous part, the closer the model is to an ordinary linear model, and vice versa. Naturally, achieving a proper balance between the two parts contributes to a better model fit.

We shall note that model (6) could be easily extended to a generalized linear model (GLM) setting to accommodate other types of outcomes. For example, by allowing certain link function $f(\cdot)$, we may consider a GLM of the form $f(\mathbb{E}(y_i)) = \langle \boldsymbol{X}_i, \boldsymbol{C}_i \rangle$. Then, the coefficients $\boldsymbol{C}_i$ could still be modeled as in (3) to (5).

To learn the coefficients $\boldsymbol{C}$, $\boldsymbol{D}$ and $\boldsymbol{W}$, we propose the following penalized minimization problem

$$\min_{\boldsymbol{C},\boldsymbol{D},\boldsymbol{W}} \quad \frac{1}{n} \sum_{i=1}^{n} \left(y_i - \langle \boldsymbol{X}_i, \boldsymbol{C}_i \rangle\right)^2 + \lambda_1 \|\boldsymbol{C}\|_F^2 + \lambda_2 \|\boldsymbol{D}\|_F^2, \tag{7}$$

$$\text{s.t.} \quad \boldsymbol{C}_i = \boldsymbol{C} + g(\boldsymbol{X}_i \boldsymbol{W} \boldsymbol{X}_i^\top)^\top \boldsymbol{D}, \ \|\boldsymbol{W}\|_F = 1,$$

where $\|\cdot\|_F$ is the Frobenius norm and $\lambda_1$ and $\lambda_2$ are regularization parameters to balance the homogeneous and heterogeneous effects. Besides, a norm constraint for $\boldsymbol{W}$ is also required due to identifiability consideration. We defer to Section 4 for the computation of (7).

## 3    Individualized regression and attention

We refer to our individualized modeling as Attention boosted Individualized Regression due to its connections with the self-attention mechanism. The self-attention mechanism was proposed in the seminal work [21], and the Transformer model based on it has demonstrated exceptional performance in natural language processing, computer vision, and other fields. In this section, we establish the connection between the proposed model (6) and the self-attention mechanism.

Given the input $\boldsymbol{X} \in \mathbb{R}^{n \times d}$, the Scaled Dot-Product Attention mechanism computes the output using $\boldsymbol{Q} \in \mathbb{R}^{n \times d_k}$, $\boldsymbol{K} \in \mathbb{R}^{n \times d_k}$, and $\boldsymbol{V} \in \mathbb{R}^{n \times d_v}$, representing query, key, and value, respectively. The three essential components are linearly transformed from $\boldsymbol{X}$ by

$$\boldsymbol{Q} = \boldsymbol{X} \boldsymbol{W}_Q, \ \boldsymbol{K} = \boldsymbol{X} \boldsymbol{W}_K, \ \boldsymbol{V} = \boldsymbol{X} \boldsymbol{W}_V$$

with corresponding weight matrices $\boldsymbol{W}_Q$, $\boldsymbol{W}_K$, and $\boldsymbol{W}_V$. Incorporating a softmax function for normalization, the Scaled Dot-Product Attention is defined as

$$f(\boldsymbol{X}) = \text{softmax}\left(\frac{\boldsymbol{Q}\boldsymbol{K}^\top}{\sqrt{d_k}}\right)\boldsymbol{V}. \tag{8}$$

In the attention mechanism, the first part $\text{softmax}\left(\frac{\boldsymbol{Q}\boldsymbol{K}^\top}{\sqrt{d_k}}\right)$ essentially computes the pairwise similarity between queries and keys, normalized by a combination of scaling and row-wise softmax. With the resulting attention map, the output is obtained by reweighing the pairs' values. The attention map is at the core of the attention mechanism, as it provides an individualized map that captures information on pairwise similarity within each sample.

Moreover, the attention mechanism (8) could also be expressed in a row-wise form. Let $\boldsymbol{O} = \text{Attention}(\boldsymbol{Q}, \boldsymbol{K}, \boldsymbol{V})$ be the output of the attention function. Further let $\boldsymbol{o}_i, \boldsymbol{q}_i, \boldsymbol{k}_i$ and $\boldsymbol{v}_i$ be the $i$-th row of $\boldsymbol{O}, \boldsymbol{Q}, \boldsymbol{K}$, and $\boldsymbol{V}$, respectively. Then, (8) is equivalent to

$$\boldsymbol{o}_i = \frac{\sum_{j=1}^{N} \exp\left(\boldsymbol{q}_i^\top \boldsymbol{k}_j / \sqrt{d_k}\right) \boldsymbol{v}_j}{\sum_{j=1}^{N} \exp\left(\boldsymbol{q}_i^\top \boldsymbol{k}_j / \sqrt{d_k}\right)}. \tag{9}$$

This form clearly highlights that the basis of the weights in the attention map is formed by vector correlation. In fact, the dot-product-based pairwise similarity is derived from a nonlinear transformation of the correlation between pairs. Beyond softmax function, normalization in attention could also be accomplished using a general function $g(\cdot)$. As a result, we obtain the following generalized attention

$$f(\boldsymbol{X}) = g\left(\boldsymbol{Q}\boldsymbol{K}^\top\right)\boldsymbol{V}. \tag{10}$$

The attention mechanism in the form of (10) with a nonlinear function $g(\cdot)$ can face computational challenges, as the direct computation of attention maps requires significant resources to handle $n \times n$ matrices. To address the computation issues, several recent works have emerged, such as sparse transformers [4], efficient transformers [13], and more. Linear attention mechanisms have been studied as a subcategory, which can dramatically decrease complexity from quadratic to linear. [16] proposed a linear attention boosted on the first-order Taylor expansion of the exponential part in the softmax function, i.e., $\exp(\boldsymbol{q}^\top \boldsymbol{k}) \approx 1 + \boldsymbol{q}^\top \boldsymbol{k}$. [12] presented the linearized attention using kernel functions, which measure the similarity between $\boldsymbol{q}$ and $\boldsymbol{k}$ through $\phi(\boldsymbol{q})^\top \phi(\boldsymbol{k})$. In this case, $\phi(\cdot)$ represents a specific kernel function. [22] introduced Linformer, which leverages the low-rankness of the attention map to reduce complexity to linear. Notably, [19] considered linear $\rho(\boldsymbol{Y}) = \boldsymbol{Y}/n$ as scaling normalization, consequently,

$$f(\boldsymbol{X}) = \frac{1}{n} \boldsymbol{Q} \boldsymbol{K}^\top \boldsymbol{V}. \tag{11}$$

Linear attention mechanisms are efficient because they bypass the need to compute $n \times n$ matrices by using associative multiplication, reducing complexity from $O(n^2)$ to $O(n)$. While on the other hand, experiments show that Linear attentions does not result in a significant compromise in performance.

Now we demonstrate the connections between our individualized regression model (6) and the self-attention mechanism. We let the homogeneous coefficient $\boldsymbol{C} = \boldsymbol{0}$ and focus on the model

$$y_i = \langle \boldsymbol{X}_i, g(\boldsymbol{X}_i \boldsymbol{W} \boldsymbol{X}_i^\top)^\top \boldsymbol{D} \rangle + \varepsilon_i. \tag{12}$$

**Proposition 3.1.** *Suppose the model (12) holds and matrices $\boldsymbol{W}$ and $\boldsymbol{D}$ in model (12) could be decomposed as below*

*(I) $\boldsymbol{W} = \boldsymbol{W}_Q \boldsymbol{W}_K^\top$ for two matrices $\boldsymbol{W}_Q, \boldsymbol{W}_K \in \mathbb{R}^{d \times d_k}$ with $d_k \leq d$,*

*(II) $\boldsymbol{D} = \boldsymbol{B} \boldsymbol{W}_V^\top$ for two matrices $\boldsymbol{B}, \boldsymbol{W}_V \in \mathbb{R}^{d \times d_v}$ with $d_v \leq d$.*

*Then, the following equation holds for each sample $\boldsymbol{X}_i$*

$$\left\langle \boldsymbol{X}_i, \ g(\boldsymbol{X}_i \boldsymbol{W} \boldsymbol{X}_i^\top)^\top \boldsymbol{D} \right\rangle = \left\langle g(\boldsymbol{Q}_i \boldsymbol{K}_i^\top) \boldsymbol{V}_i, \ \boldsymbol{B} \right\rangle, \tag{13}$$

*where*

$$\boldsymbol{Q}_i = \boldsymbol{X}_i \boldsymbol{W}_Q, \ \ \boldsymbol{K}_i = \boldsymbol{X}_i \boldsymbol{W}_K, \ \ \boldsymbol{V}_i = \boldsymbol{X}_i \boldsymbol{W}_V.$$

*Remark* 3.2. Proposition 3.1 establishes the connection between our individualized regression model (12) and the self-attention mechanism (10). We shall note that the product of the query and key $\boldsymbol{Q}_i \boldsymbol{K}_i^\top = \boldsymbol{X}_i \boldsymbol{W} \boldsymbol{X}_i^\top$ essentially acts as an internal relation map, capturing the inter-dependence between local patches. By applying an appropriate function $g(\cdot)$, we can obtain the normalized sample-specific internal relation map. Furthermore, the value matrix $\boldsymbol{V}_i$ can be enhanced by multiplying with such relation map. The final outcome is obtained as the inner product of the aggregated features and the coefficient matrix.

It is important to note that the two conditions in the proposition are mild, as they correspond to the low-rank assumptions: (I) rank$(\boldsymbol{W}) \leq d_k$ and (II) rank$(\boldsymbol{D}) \leq d_v$. In particular, (I) is consistent with Theorem 1 in [22], which demonstrated that the self-attention mechanism, i.e., the attention matrix, is low-rank. Moreover, if assumption (II) is not considered, (13) becomes equivalent to the simplified Vision Transformer in [11] without considering the value.

Conversely, the equivalence (13) also helps understand our model from the perspective of the self-attention mechanism. Since the tuple $(\boldsymbol{W}_Q, \boldsymbol{W}_K, \boldsymbol{W}_V)$ represents embedding projections in self-attention, $\boldsymbol{W} = \boldsymbol{W}_Q \boldsymbol{W}_K^\top$ is equivalent to a composite embedding that is adaptive and determines the attention map. Meanwhile, $\boldsymbol{D} = \boldsymbol{B} \boldsymbol{W}_V^\top$ represents a projected regression coefficient that can be learned as a whole. The heterogeneous coefficients $\boldsymbol{D}_i = g(\boldsymbol{X}_i \boldsymbol{W} \boldsymbol{X}_i^\top)^\top \boldsymbol{D}$ can be considered as an aggregation of base coefficients through the attention map, contributing to model interpretation. If we set $g(\cdot)$ as the identity function, (13) simplifies to linear attention, thus enjoying the computational advantages of linear attention.

We should also note that with the growing popularity of transformers in natural language processing, self-attention-based architectures have begun to be introduced in computer vision, encompassing

various visual tasks such as detection, segmentation, and generation. However, we mainly discuss their initial involvement in regression and classification tasks [23, 3, 5]. In particular, [5] directly applied a pure transformer to address the image classification problem and proposed Vision Transformer (ViT). ViT treats images as sequences by dividing them into fixed-size patches and processes them using a transformer architecture. ViT comprises two main components: the Encoder and the Classifier. In the transformer encoder, each attention map is computed for each image based on patch-wise similarity. The embedded patches are then followed by a multilayer perceptron head that serves as a regressor/classifier. Although some details are not discussed, this simplification helps to understand the connection with our model. More recently, [11] proposed simplifying transformer blocks. By removing skip connections, value parameters, sequential sub-blocks, and normalization layers, the simplified transformer has the potential to achieve fewer parameters and faster training.

## 4 Computation

In this section, we demonstrate the computation of the penalized minimization problem (7). From now on, we shall focus on the special case where $g(x) = x$ is the identity function, which corresponds to linear attention. Namely, the model is

$$y_i = \langle \boldsymbol{X}_i, \boldsymbol{C} + \boldsymbol{X}_i \boldsymbol{W}^\top \boldsymbol{X}_i^\top \boldsymbol{D} \rangle + \varepsilon_i. \tag{14}$$

In this context, we develop an alternating minimization algorithm and highlight its benefits compared to gradient-based ones. First, we observe that the heterogeneous part in model (14) satisfies

$$\left\langle \boldsymbol{X}_i, \boldsymbol{X}_i \boldsymbol{W}^\top \boldsymbol{X}_i^\top \boldsymbol{D} \right\rangle = \left\langle \boldsymbol{X}_i^\top \boldsymbol{D} \boldsymbol{X}_i^\top \boldsymbol{X}_i, \boldsymbol{W} \right\rangle = \left\langle \boldsymbol{X}_i \boldsymbol{W} \boldsymbol{X}_i^\top \boldsymbol{X}_i, \boldsymbol{D} \right\rangle. \tag{15}$$

Moreover, let $\boldsymbol{w} = \text{vec}(\boldsymbol{W})$ and $\boldsymbol{d} = \text{vec}(\boldsymbol{D})$ be the vectorization of $\boldsymbol{W}$ and $\boldsymbol{D}$. It holds that

$$\left\langle \boldsymbol{X}_i^\top \boldsymbol{D} \boldsymbol{X}_i^\top \boldsymbol{X}_i, \boldsymbol{W} \right\rangle = \left\langle \boldsymbol{Z}_i, \boldsymbol{w} \boldsymbol{d}^\top \right\rangle, \quad \text{where } \boldsymbol{Z}_i = \left( \boldsymbol{X}_i^\top \boldsymbol{X}_i \right) \otimes \boldsymbol{X}_i^\top \tag{16}$$

and $\otimes$ denotes the Kronecker product. Clearly, (16) displays a bilinear form. We start our algorithm by initializing $\boldsymbol{w}$ as the top left singular vector of $\sum_{i=1}^n y_i \boldsymbol{Z}_i$. Formally,

$$\widehat{\boldsymbol{w}}^{(0)} = \text{SVD}_u \left( \sum_{i=1}^n y_i \boldsymbol{Z}_i \right), \tag{17}$$

where $\text{SVD}_u(\cdot)$ represents the top left singular vector of a matrix.

Now we introduce our alternating minimization algorithm. Denote $\widehat{\boldsymbol{C}}^{(t)}$, $\widehat{\boldsymbol{D}}^{(t)}$ and $\widehat{\boldsymbol{W}}^{(t)}$ as the iterates in $t$-th loop. According to (15), we alternatively update $\left( \widehat{\boldsymbol{C}}^{(t)}, \widehat{\boldsymbol{D}}^{(t)} \right)$ and $\widehat{\boldsymbol{W}}^{(t)}$ as below. Given $\widehat{\boldsymbol{W}}^{(t-1)}$, denote $\boldsymbol{U}_i^{(t-1)} = \boldsymbol{X}_i \widehat{\boldsymbol{W}}^{(t-1)} \boldsymbol{X}_i^\top \boldsymbol{X}_i$. Then $\left( \widehat{\boldsymbol{C}}^{(t)}, \widehat{\boldsymbol{D}}^{(t)} \right)$ can be updated by

$$\left( \widehat{\boldsymbol{C}}^{(t)}, \widehat{\boldsymbol{D}}^{(t)} \right) = \underset{\boldsymbol{C}, \boldsymbol{D}}{\text{argmin}} \frac{1}{n} \sum_{i=1}^n \left( y_i - \left\langle \left[ \boldsymbol{X}_i, \boldsymbol{U}_i^{(t-1)} \right], \left[ \boldsymbol{C}, \boldsymbol{D} \right] \right\rangle \right)^2 + \lambda_1 \|\boldsymbol{C}\|_F^2 + \lambda_2 \|\boldsymbol{D}\|_F^2. \tag{18}$$

Clearly, (18) can be seen as a ridge-like regression with two levels of penalization on distinct coefficients, which has an explicit solution shown in Section A in the appendix.

Given $\left( \widehat{\boldsymbol{C}}^{(t)}, \widehat{\boldsymbol{D}}^{(t)} \right)$, then $\widehat{\boldsymbol{W}}^{(t)}$ can be updated by

$$\widehat{\boldsymbol{W}}^{(t)} = \underset{\boldsymbol{W}}{\text{argmin}} \frac{1}{n} \sum_{i=1}^n \left( y_i - \left\langle \boldsymbol{X}_i, \widehat{\boldsymbol{C}}^{(t)} \right\rangle - \left\langle \boldsymbol{X}_i^\top \widehat{\boldsymbol{D}}^{(t)} \boldsymbol{X}_i^\top \boldsymbol{X}_i, \boldsymbol{W} \right\rangle \right)^2, \tag{19}$$

$$\widehat{\boldsymbol{W}}^{(t)} = \widehat{\boldsymbol{W}}^{(t)} / \|\widehat{\boldsymbol{W}}^{(t)}\|_F. \tag{20}$$

It implies that $\widehat{\boldsymbol{W}}^{(t)}$ could be obtained easily through ordinary least squares followed by normalization. We summarize the alternating minimization algorithm in Algorithm 1 in Section A in the appendix. In practice, the regularization level $(\lambda_1, \lambda_2)$ are treated as hyperparameters and we can use cross-validation to search for the optimal combination.

# 5 Theoretical analysis

In this section, we provide theoretical guarantees for our Attention boosted Individualized Regression. Specifically, we show that $\boldsymbol{W}^{(t)}$ and $\boldsymbol{D}^{(t)}$ obtained by alternating minimization algorithm converge to the true counterparts at a geometric rate. To simplify analysis, we focus on the heterogeneous part of model (14), although our results can be extended to more general cases. Suppose that

$$y_i = \langle \boldsymbol{X}_i, \boldsymbol{X}_i \boldsymbol{W}^\top \boldsymbol{X}_i^\top \boldsymbol{D} \rangle + \varepsilon_i. \tag{21}$$

Let $\boldsymbol{w} = \mathrm{vec}(\boldsymbol{W})$ and $\boldsymbol{d} = \mathrm{vec}(\boldsymbol{D})$, the optimization problem could be written as

$$\min_{\boldsymbol{d}, \boldsymbol{w}} \frac{1}{n} \sum_{i=1}^n \left\{ y_i - \left\langle \left( \boldsymbol{X}_i^\top \boldsymbol{X}_i \right) \otimes \boldsymbol{X}_i^\top, \boldsymbol{w} \boldsymbol{d}^\top \right\rangle \right\}^2 + \lambda_2 \|\boldsymbol{d}\|_2^2. \tag{22}$$

which is non-convex on $\boldsymbol{w}$ and $\boldsymbol{d}$. For the rearranged images $\boldsymbol{X}_i$ for $i = 1, \ldots, n$, we define

$$\boldsymbol{Z} = \left( \mathrm{vec} \left\{ \left( \boldsymbol{X}_1^\top \boldsymbol{X}_1 \right) \otimes \boldsymbol{X}_1^\top \right\}, \ldots, \mathrm{vec} \left\{ \left( \boldsymbol{X}_n^\top \boldsymbol{X}_n \right) \otimes \boldsymbol{X}_n^\top \right\} \right)^\top. \tag{23}$$

Here each row of $\boldsymbol{Z}$ represents a transformed sample. For the new feature matrix $\boldsymbol{Z}$, we suppose the following RIP condition.

*Condition 5.1. (Restricted Isometry Property)* For each integer $= 1, 2, \ldots$, a matrix $\boldsymbol{P} \in \mathbb{R}^{n \times q_1 q_2}$ is said to satisfy the $r$-RIP condition with constant $\delta_r \in (0, 1)$, if for all $\boldsymbol{M} \in \mathbb{R}^{q_1 \times q_2}$ of rank at most $r$, it holds that

$$(1 - \delta_r) \|\boldsymbol{M}\|_F^2 \leq 1/n \|\boldsymbol{P} \mathrm{vec}(\boldsymbol{M})\|_2^2 \leq (1 + \delta_r) \|\boldsymbol{M}\|_F^2. \tag{24}$$

The Restricted Isometry Property (RIP) was initially introduced by [2] for sparse vector recovery and subsequently extended by [17] for low-rank matrices, as in Condition 5.1. Many random matrices with an adequately large number of independent observations, such as Gaussian or sub-Gaussian matrices, satisfy the RIP condition [17]. In our analysis, we require that $\boldsymbol{Z}$ defined in (23) satisfies the 2-RIP condition with constant $\delta_2$.

To evaluate the estimation error of parameters, we consider an angle-based distance between two matrices. Formally, for any two matrices $\boldsymbol{U}$ and $\boldsymbol{V}$ with the same dimension, we define the distance as $\mathrm{dist}(\boldsymbol{U}, \boldsymbol{V}) = \sqrt{1 - \langle \boldsymbol{U}, \boldsymbol{V} \rangle^2 / (\|\boldsymbol{U}\|_F^2 \|\boldsymbol{V}\|_F^2)}$. This distance metric corresponds to the squared sine value after vectorization, that is, $\mathrm{dist}(\boldsymbol{U}, \boldsymbol{V}) = \sin(\boldsymbol{u}, \boldsymbol{v})$, where $\boldsymbol{u} = \mathrm{vec}(\boldsymbol{U})$ and $\boldsymbol{v} = \mathrm{vec}(\boldsymbol{V})$. Now we are ready to present our main theorem.

**Theorem 5.2.** *Suppose model (21) holds and solved by alternating minimization algorithm. Assume that $\boldsymbol{Z}$ satisfies 2-RIP Condition 5.1 with a constant $\delta_2$. Denote $\mu_0 = \mathrm{dist}\left( \widehat{\boldsymbol{W}}^{(0)}, \boldsymbol{W} \right)$ as the initial distance. Let $\kappa_1 = \mu_0/2 + 3\delta_2/(1 - 3\delta_2)$ and $\kappa_2 = \mu_0/2 + (3\delta_2 + \lambda_2)/(1 - 3\delta_2 + \lambda_2)$ and assume $\kappa_1, \kappa_2 < 1$. And $\tau_1, \tau_2$ are noise related terms. Suppose $\kappa_1 \mu_0 + \tau_1 \leq \mu_0$ and $\kappa_2 \mu_0 + \tau_2 \leq \mu_0$. Then, after $t$ iterations we have*

$$\mathrm{dist}\left( \widehat{\boldsymbol{W}}^{(t)}, \boldsymbol{W} \right) \leq (\kappa_1 \kappa_2)^t \mu_0 + \frac{\kappa_1 \tau_2 + \tau_1}{1 - \kappa_1 \kappa_2}, \tag{25}$$

$$\mathrm{dist}\left( \widehat{\boldsymbol{D}}^{(t)}, \boldsymbol{D} \right) \leq \kappa_1^{t-1} \kappa_2^t \mu_0 + \frac{\kappa_2 \tau_1 + \tau_2}{1 - \kappa_1 \kappa_2}. \tag{26}$$

Theorem 5.2 suggests that the estimation errors of $\boldsymbol{W}^{(t)}$ and $\boldsymbol{D}^{(t)}$ converge at a geometric rate, with the contraction parameter being $\kappa_1 \kappa_2$. On the right-hand-side of (25) and (26), the first term represents the optimization error, while the second term represents the statistical error. It becomes evident that the optimization error decays geometrically with each iteration $t$.

**Theorem 5.3.** *Suppose model (21) holds and solved by alternating minimization algorithm. Assume that $\boldsymbol{Z}$ satisfies 2-RIP Condition 5.1 with a constant $\delta_2$. Denote $\mu_0 = \|\widehat{\boldsymbol{W}}^{(0)} - \boldsymbol{W}\|_F$ as the initialization error. Let $\nu_1 = 2\mu_0 + 3\delta_2/(1 - 3\delta_2)$ and $\nu_2 = 2\mu_0 + (3\delta_2 + \lambda_2)/(1 - 3\delta_2 + \lambda_2)$, and assume $\nu_1, \nu_2 < 1$. And $\tau_1, \tau_2$ are noise related terms. Suppose $\nu_1 \mu_0 + \tau_1 \leq \mu_0$ and $\nu_2 \mu_0 + \tau_2 \leq \mu_0$. Then, after $t$ iterations we have*

$$\|\widehat{\boldsymbol{Y}}^{(t)} - \boldsymbol{Y}\|_2 \leq 3\|\boldsymbol{D}\|_F \sqrt{1 + \delta_2} \left\{ (\nu_1 \nu_2)^{t-1} \mu_0 + \frac{\tau_1 + \tau_2}{1 - \nu_1 \nu_2} \right\}. \tag{27}$$

Theorem 5.3 suggests that the prediction error decreases in a similar manner as the estimation errors in Theorem 5.2. It is important to note that the error bounds in both theorems are dependent on suitable initialization. We employ spectral initialization as shown in (17), which has been proven to have an error closely approximating the true value.

# 6 Simulation

We conduct extensive simulation studies to evaluate the performance of our Attention boosted Individualized Regression compared to related methods in this section. Besides, ablation studies are deferred to Section B.1 in the appendix to show the advantage of combining the homogeneous and heterogeneous parts. Throughout the simulation, we assume that the data is generated according to the model (6). The size of the images is set to $28 \times 28$, with a sample size of 4000 for training and 1000 for testing. The noise $\varepsilon_i$ follows an i.i.d. $\mathcal{N}(0, 1)$. The coefficient matrices $\boldsymbol{C}^{\mathrm{ori}}$ and $\boldsymbol{D}^{\mathrm{ori}}$ are generated as two circles depicted in Figure 1. For the images $\boldsymbol{X}_i^{\mathrm{ori}}$, we assume that internal relations exist among blocks of size $4 \times 4$ within each image, where two blocks at random locations are correlated. The entries in $\boldsymbol{X}_i^{\mathrm{ori}}$ follow i.i.d $\mathcal{N}(0, 1)$, while the correlated blocks are generated using the two methods below.

Case 1: With specific $\boldsymbol{W}$, the internal relations are subject to (5). Consider a low-rank $\boldsymbol{W} = 2 \cdot \boldsymbol{u}_1 \boldsymbol{v}_1^\top + 1 \cdot \boldsymbol{u}_2 \boldsymbol{v}_2^\top$ where $\boldsymbol{u}_1, \boldsymbol{u}_2$ and $\boldsymbol{v}_1, \boldsymbol{v}_2$ are random vectors with entries subject to i.i.d. $\mathcal{N}(0, 1)$. Then, the correlated blocks are generated as $\boldsymbol{u}_1$ plus noise vectors with i.i.d. entries from $\mathcal{N}(0, 0.25)$.

Case 2: Without specific $\boldsymbol{W}$, the internal relations are Pearson correlation coefficients. Given a random vector $\boldsymbol{u}$ as a base with i.i.d. entries from $\mathcal{N}(0, 1)$, the correlated blocks are also generated as $\boldsymbol{u}$ plus noise vectors with i.i.d. entries from $\mathcal{N}(0, 0.25)$. Then $\boldsymbol{A}_i$ is taken as the correlation matrix where $(j, k)$-th element of $\boldsymbol{A}_i$ is the Pearson correlation coefficient between $j$-th and $k$-th blocks within $\boldsymbol{X}_i^{\mathrm{ori}}$.

Furthermore, we consider different levels of model individualization and investigate the effects on model performance. To this end, we define the degree of individuation (DI) of model (6) by the relative total magnitude of the heterogeneous part and homogeneous part. Specifically, $\mathrm{DI} = \sqrt{\sum_{i=1}^n \langle \boldsymbol{X}_i, \boldsymbol{D}_i \rangle^2 / \sum_{i=1}^n \langle \boldsymbol{X}_i, \boldsymbol{C} \rangle^2}$.

The performance of AIR is compared with four competing methods, including, low-rank matrix regression [LRMR, 27], tensor regression with lasso penalty [TRLasso, 28], Deep Kronecker Network [DKN, 7], and Vision Transformer [ViT, 5], respectively. Implementation details are provided in Section B.2 in the appendix. Of note is that we cannot implement several individualized regression methods [25, 14] as they require additional information of unknown variables. We evaluate prediction performance of different methods, measured by the root mean squared error (RMSE) on test set: $\sqrt{(1/n_{\mathrm{test}}) \sum_{i=1}^{n_{\mathrm{test}}} (\hat{y}_i^{\mathrm{test}} - y_i^{\mathrm{test}})^2}$. The average and standard error of 100 repetitions are reported in Table 1, and the estimated coefficients of different methods are illustrated in Figure 1 and 4.

Table 1: Prediction errors of different methods.

| | DI | Methods | | | | |
|---|---|---|---|---|---|---|
| | | AIR | LRMR | TRLasso | DKN | ViT |
| | 0.5 | 4.422 (0.130) | 6.616 (0.020) | 8.215 (0.021) | 4.886 (0.018) | 18.429 (0.049) |
| Case 1 | 1.0 | 8.102 (0.325) | 13.101 (0.040) | 14.655 (0.044) | 7.028 (0.032) | 18.351 (0.047) |
| | 2.0 | 10.599 (0.816) | 26.239 (0.081) | 27.007 (0.085) | 11.741 (0.043) | 24.098 (0.069) |
| | 0.5 | 3.590 (0.046) | 6.766 (0.018) | 8.337 (0.021) | 8.269 (0.018) | 24.492 (0.063) |
| Case 2 | 1.0 | 6.632 (0.022) | 13.408 (0.037) | 14.739 (0.039) | 14.964 (0.034) | 29.939 (0.084) |
| | 2.0 | 13.002 (0.044) | 26.864 (0.074) | 27.484 (0.073) | 28.686 (0.060) | 44.036 (0.111) |

The numerical results indicate that AIR outperforms all other methods, with the advantage increasing as the degree of individuation becomes greater. Figure 1 and 4 demonstrate that AIR, when solved by our algorithm, can accurately recover the shape of the true parameters. It is worth noting that in Case 2, even though our model is mis-specified with no explicit $\boldsymbol{W}$ exists, AIR still performs well. Common-model methods such as LRMR, TRLasso, and DKN tend to estimate the sum of the true coefficients for both parts. On the other hand, ViT typically requires a large number of samples and is thus not as effective due to the limited sample size.

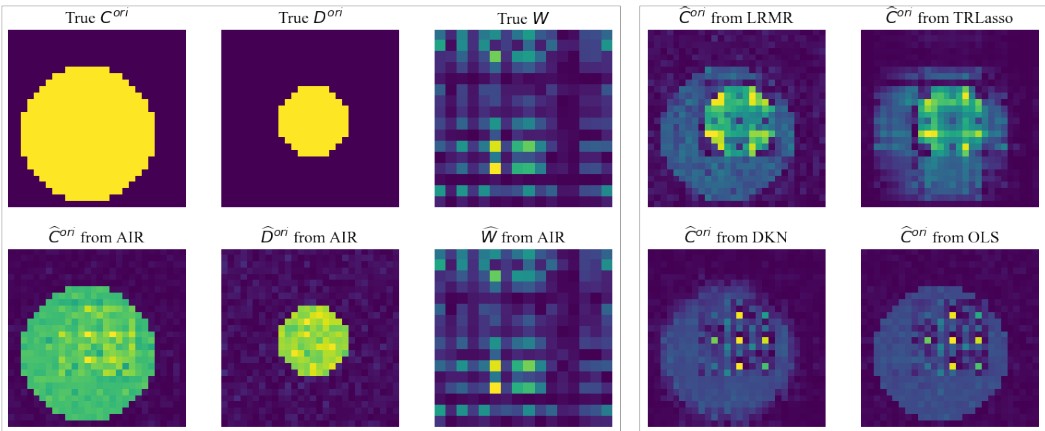

Figure 1: Case 1 simulation results with DI $= 1.0$. The first three columns show true parameters and estimations from AIR. The last two columns show estimations from other methods except ViT, as it has no explicit coefficient matrix. An additional OLS estimation is added for reference.

# 7    Real data analysis

In this section, we analyze the relationship between cognitive assessment scores and brain MRI data from the Alzheimer's Disease Neuroimaging Initiative (ADNI). The ADNI is a study on Alzheimer's disease (AD) that includes clinical, genetic, and imaging data, covering AD patients, individuals with mild cognitive impairment (MCI), and healthy controls. We collected a total of 1059 subjects from ADNI 1 and GO/2 phases with Mini-Mental State Examination (MMSE) score and brain MRI. The MMSE score measures a patient's cognitive impairment which can assist in the diagnosis of AD. Brain MRI were carefully preprocessed following a standard pipeline involving denoising, registration, skull-stripping and so on and were resized to tensors of size $48 \times 60 \times 48$ for computation efficiency. Then we extracted 10 middle coronal slices for each subject, resulting in images of size $48 \times 48$. Two samples are shown in the first column in Figure 2.

We compare the methods described in the simulation section by 5-fold cross-validation in test RMSE, of which average and standard error are presented in Table 2. AIR exhibits the best prediction performance among all methods, of which the significance can be shown by paired t-test. Furthermore, Figure 2 compares estimations of different methods while illustrates the individualized estimations from AIR for two different subjects, including the heterogeneous effect $\widehat{D}^{\text{ori}}$ and significant internal relations. To screen significant internal relations for each subject, we summarize relations of each node in the internal relation matrix $\widehat{A}_i$ and select top 5 as significant nodes. Subsequently, we mark these nodes at corresponding locations in the original sample by red boxes and show their relations by a chord diagram. For example, the block $(4, 5)$ in sample 1 has the strongest relations, and is related to both $(6, 4)$ and $(6, 5)$, indicating the important relations between corpus callosum and hippocampus. We also note that after separating heterogeneous effect, the homogeneous effect $\widehat{C}^{\text{ori}}$ highlights regions of the hippocampus, which have been acknowledged in medical literature as a crucial substructure associated with Alzheimer's disease [1]. By this means, we can find important regions and relations among them for each subject, which is potential to help personalized treatment. In contrast, other methods do not reveal clear shapes and fail to offer valuable interpretations.

Table 2: Prediction errors of different methods.

| AIR | LRMR | TRLasso | DKN | ViT |
|---|---|---|---|---|
| **3.145 (0.019)** | 3.715 (0.008) | 3.292 (0.023) | 3.261 (0.017) | 3.282 (0.025) |

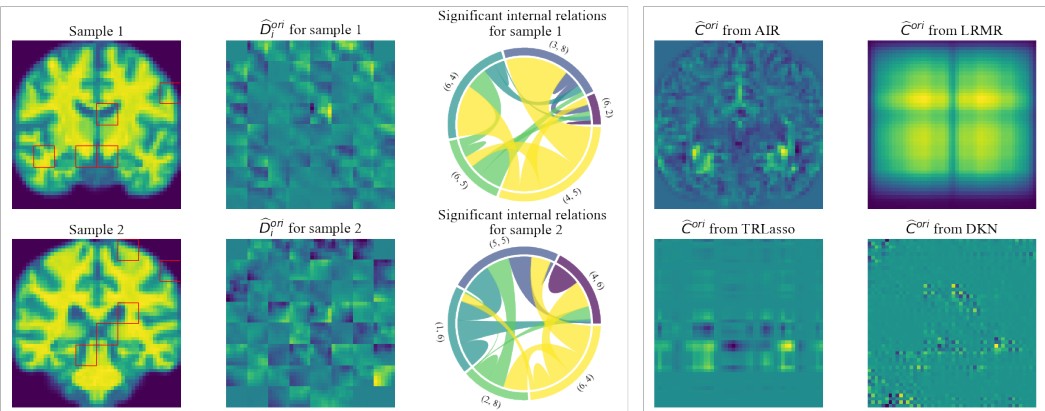

Figure 2: Results on ADNI dataset. (I) Column 1 shows two original samples. Column 2 shows heterogeneous coefficients estimated by AIR. Column 3 presents chord diagrams that illustrate the significant internal relations estimated by AIR. Each coordinate in the chord diagram corresponds to a red box marked in the sample. (II) Columns 4 and 5 compare the homogeneous coefficients estimated by AIR with the coefficients obtained from other methods.

## 8   Discussion

In this paper, we present an Attention boosted Individualized Regression model that emphasizes internal relationships within samples and is based on the concept of rotation vector correlation. Our method is specifically tailored for data with heterogeneous internal relationships. By concentrating on the internal relations within samples, our approach effectively addresses the complex and heterogeneous nature of data, making it highly beneficial for various fields, particularly, brain imaging analysis and personalized medicine. On the other hand, we realize that the AIR framework also has limitations. First, its capability to handle general data is more or less restricted. When there are minimal heterogeneous effects, its performance will be similar to an ordinary linear model. Second, as discussed earlier, our framework could be viewed as a simplified version of the Vision Transformer; however, such simplifications may also reduce its approximation power for more complex scenarios. Furthermore, this paper primarily investigates the linear form of AIR. Although the linear form performs well in the cases of interest, it remains worthwhile to explore the generalization of the model in future work.

### Acknowledgments and Disclosure of Funding

We thank the anonymous reviewers for their helpful comments. Yuan Cao is supported by NSFC 12301657 and Hong Kong RGC grant ECS 27308624. Long Feng is supported by Hong Kong RGC grant GRF 17301123 and ECS 21313922.

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

# Appendix

## A   Computation

The pseudocode of the alternating minimization algorithm is summarized in Algorithm 1. As mentioned in Section 4, updating $\left(\widehat{\boldsymbol{C}}^{(t)}, \widehat{\boldsymbol{D}}^{(t)}\right)$ is a ridge-like regression problem and updating $\widehat{\boldsymbol{W}}^{(t)}$ is an ordinary least squares problem, both of which have explicit solutions. For the former, we consider the vectorzied version of the problem (32). With bold lowercase letters being the vectorization of corresponding matrices, we have

$$\begin{pmatrix} \widehat{\boldsymbol{c}} \\ \widehat{\boldsymbol{d}} \end{pmatrix} = \operatorname*{argmin}_{\boldsymbol{c}, \boldsymbol{d}} \frac{1}{n} \sum_{i=1}^{n} \left\{ y_i - \left( \boldsymbol{x}_i^\top, \boldsymbol{u}_i^\top \right) \begin{pmatrix} \boldsymbol{c} \\ \boldsymbol{d} \end{pmatrix} \right\}^2 + \lambda_1 \|\boldsymbol{c}\|_2^2 + \lambda_2 \|\boldsymbol{d}\|_2^2 \tag{28}$$

$$= \operatorname*{argmin}_{\boldsymbol{\beta}} \|\boldsymbol{Y} - \boldsymbol{N}\boldsymbol{\beta}\|_2^2 + \boldsymbol{\beta}^\top \boldsymbol{\Lambda} \boldsymbol{\beta}, \tag{29}$$

where $\boldsymbol{\beta}$ stores all coefficients, $\boldsymbol{N}$ is the new design matrix within this step and $\boldsymbol{\Lambda} = \begin{pmatrix} \lambda_1 \boldsymbol{I} & \boldsymbol{0} \\ \boldsymbol{0} & \lambda_2 \boldsymbol{I} \end{pmatrix}$ includes different intensities of penalization. Therefore, (29) has the following solution

$$\widehat{\boldsymbol{\beta}} = \left( \boldsymbol{N}^\top \boldsymbol{N} + \boldsymbol{\Lambda} \right)^{-1} \boldsymbol{N}^\top \boldsymbol{Y}. \tag{30}$$

Similarly, the vectorization of (34) implies its OLS solution as below

$$\widehat{\boldsymbol{w}} = \operatorname*{argmin}_{\boldsymbol{w}} \left\| \widetilde{\boldsymbol{Y}} - \boldsymbol{M}\boldsymbol{w} \right\|_2^2 = \left( \boldsymbol{M}^\top \boldsymbol{M} \right)^{-1} \boldsymbol{M}^\top \widetilde{\boldsymbol{Y}}, \tag{31}$$

where $\widetilde{\boldsymbol{Y}}$ is the response minus homogeneous part and $\boldsymbol{M}$ is the new design matrix within this step.

---

**Algorithm 1** Alternating minimization algorithm

---

**Input:** $\boldsymbol{X}_i$, $y_i$, $i = 1, \ldots, n$.
Initialize $\widehat{\boldsymbol{w}}^{(0)} = \mathrm{SVD}_u \left( \sum_{i=1}^{n} y_i \boldsymbol{Z}_i \right)$.
**repeat**

$$\left( \widehat{\boldsymbol{C}}^{(t)}, \widehat{\boldsymbol{D}}^{(t)} \right) = \operatorname*{argmin}_{\boldsymbol{C}, \boldsymbol{D}} \frac{1}{n} \sum_{i=1}^{n} \left( y_i - \left\langle \left[ \boldsymbol{X}_i, \boldsymbol{U}_i^{(t-1)} \right], \left[ \boldsymbol{C}, \boldsymbol{D} \right] \right\rangle \right)^2 + \lambda_1 \|\boldsymbol{C}\|_F^2 + \lambda_2 \|\boldsymbol{D}\|_F^2. \tag{32}$$

$$\widehat{\boldsymbol{W}}^{(t)} = \operatorname*{argmin}_{\boldsymbol{W}} \frac{1}{n} \sum_{i=1}^{n} \left( y_i - \left\langle \boldsymbol{X}_i, \widehat{\boldsymbol{C}}^{(t)} \right\rangle - \left\langle \boldsymbol{X}_i^\top \widehat{\boldsymbol{D}}^{(t)} \boldsymbol{X}_i^\top \boldsymbol{X}_i, \boldsymbol{W} \right\rangle \right)^2. \tag{33}$$

$$\widehat{\boldsymbol{W}}^{(t)} = \widehat{\boldsymbol{W}}^{(t)} / \|\widehat{\boldsymbol{W}}^{(t)}\|_F. \tag{34}$$

**until** Converges or reaches maximal iterations.
**Output:** $\widehat{\boldsymbol{C}}^{(T)}, \widehat{\boldsymbol{D}}^{(T)}, \widehat{\boldsymbol{W}}^{(T)}$.

---

## B   Experimental extras

### B.1   Ablation studies

We conduct ablation studies in this section to investigate the effects of homogeneous part and heterogeneous part. Specifically, we compare

(1) AIR: $y_i = \langle \boldsymbol{X}_i, \boldsymbol{C} \rangle + \langle \boldsymbol{X}_i, \boldsymbol{D}_i \rangle + \varepsilon_i$, subject to $\boldsymbol{D}_i = \boldsymbol{X}_i \boldsymbol{W} \boldsymbol{X}_i^\top \boldsymbol{D}$.

(2) Hetero: $y_i = \langle \boldsymbol{X}_i, \boldsymbol{D}_i \rangle + \varepsilon_i$, subject to $\boldsymbol{D}_i = \boldsymbol{X}_i \boldsymbol{W} \boldsymbol{X}_i^\top \boldsymbol{D}$.

(3) Homo: $y_i = \langle \boldsymbol{X}_i, \boldsymbol{C} \rangle + \varepsilon_i$.

Hetero refers to the AIR with only heterogeneous part, which is solved by alternately updating $\boldsymbol{D}$ and $\boldsymbol{W}$. Homo refers to the AIR with only homogeneous part which is actually a linear regression model and can be solved by OLS directly. For comparison among these three models, we follow Case 1 and Case 2 in the simulation part, i.e. with and without explicit $\boldsymbol{W}$ when generating true internal relation matrices. We extend the degree of individuation (DI) to $\{1/4, 1/2, 1, 2, 4\}$, indicating the true model becomes more and more individualized. We plot the average prediction errors, i.e. RMSE on test set, based on 100 repetitions, against DI in Figure 3 Both subplots show that the Homo is better than Hetero when DI is small while get worse when DI increases. However, the AIR is always the best all over different DI. It demonstrates the advantage of combining the homogeneous and heterogeneous parts, which adapts the model to more scenarios.

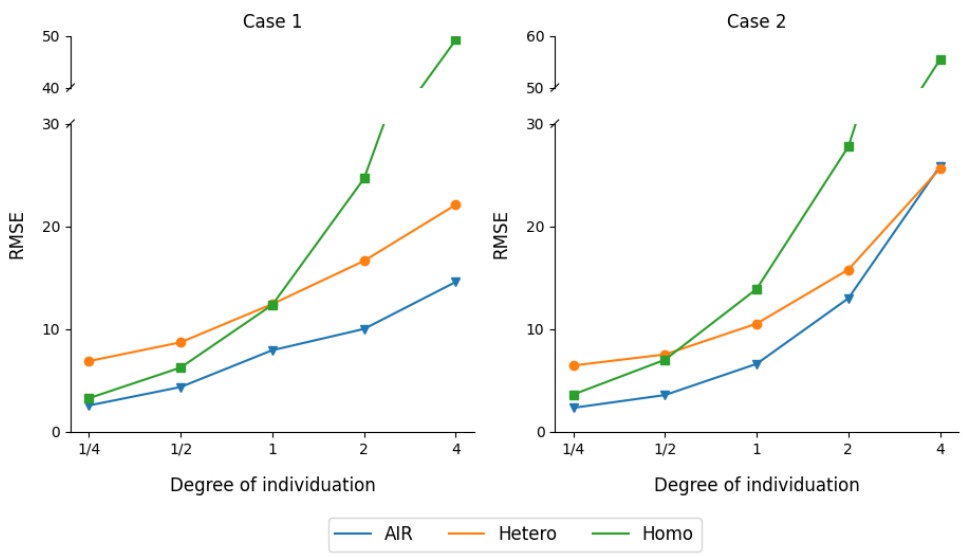

Figure 3: Results of ablation studies. Incorporating homogeneous part and heterogeneous part makes the AIR more robust, especially better than the one with only heterogeneous part.

## B.2 Simulation

Codes of our approach are available at `https://github.com/YLKnight/AIR`. Implementation details of different methods are explained here. The AIR is implemented in *Python* with hyperparameters $\lambda_1$ and $\lambda_2$ selected by 5-fold cross-validation, of which the candidate sets are both from 1 to 10. LRMR and TRLasso are implemented by their *Matlab* code, with hyperparameters selected by BIC in default setting. DKN is implemented by its *Python* code. The blocksizes are set as $2 \times 2$, $2 \times 2$ and $7 \times 7$, resulting in 3 layers while the rank is by default selected by BIC from 1 to 5. The ViT is trained by Adam optimizer in *Pytorch*. Followed by an MLP for regression, the transformer model includes 4 transformer blocks with 8 heads in each Multi-head Attention layer, and the patch size is set as $4 \times 4$. In all experiments, the CPUs used are Intel Xeon Gold 5218R and GPUs used are NVIDIA GeForce RTX 3090. Figure 4 below shows simulation results under Case 2.

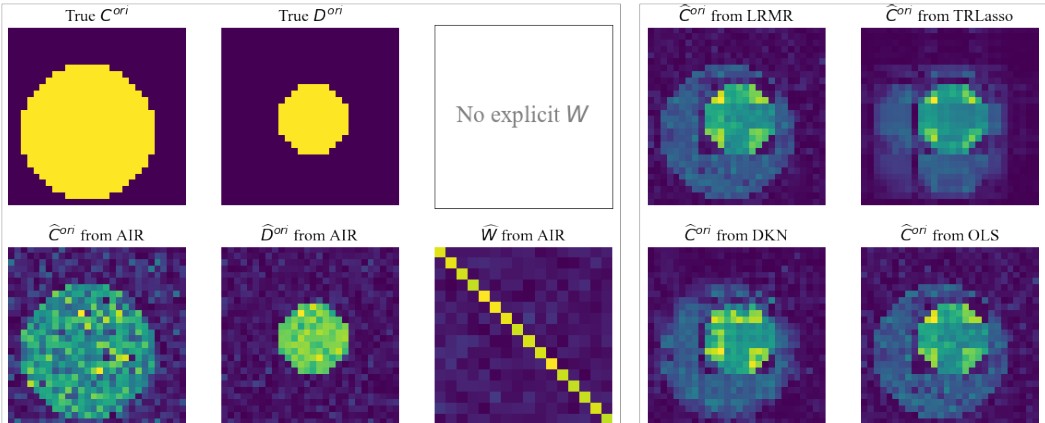

Figure 4: Simulation results under Case 2 with DI = 1.0. There does not exist an explicit true $\boldsymbol{W}$ while the internal relation matrix $\boldsymbol{A}_i$ is computed directly by patchwise Pearson correlation coefficients. Because such $\boldsymbol{A}_i$ is close to a diagonal matrix, it is rational that $\widehat{\boldsymbol{W}}$ from AIR is close to a diagonal matrix.

## C Proofs

### C.1 Useful lemmas

**Lemma C.1.** *Define the distance of two vectors $\boldsymbol{u}, \boldsymbol{v} \in \mathbb{R}^p$ as*

$$dist(\boldsymbol{u}, \boldsymbol{v}) = \sqrt{1 - \frac{\langle \boldsymbol{u}, \boldsymbol{v} \rangle^2}{\|\boldsymbol{u}\|_2^2 \|\boldsymbol{v}\|_2^2}} \tag{35}$$

*For any vectors $\boldsymbol{u}, \boldsymbol{v} \in \mathbb{R}^p$ where $\|\boldsymbol{v}\|_2 = 1$, it holds that*

$$dist\left(\boldsymbol{u}, \boldsymbol{v}\right) \leq \|\boldsymbol{u} - \boldsymbol{v}\|_2 \tag{36}$$

**Lemma C.2.** *For any vectors $\boldsymbol{u}, \boldsymbol{v} \in \mathbb{R}^p$, it holds that*

$$\|\boldsymbol{u} - \boldsymbol{v}\|_2 \geq \frac{1}{2} \left(\|\boldsymbol{u}\|_2 + \|\boldsymbol{v}\|_2\right) \left\| \frac{\boldsymbol{u}}{\|\boldsymbol{u}\|_2} - \frac{\boldsymbol{v}}{\|\boldsymbol{v}\|_2} \right\|_2 \tag{37}$$

**Lemma C.3.** *Define*

$$\boldsymbol{Z} = \left(vec\left(\left(\boldsymbol{X}_1^\top \boldsymbol{X}_1\right) \otimes \boldsymbol{X}_1^\top\right), \ldots, vec\left(\left(\boldsymbol{X}_n^\top \boldsymbol{X}_n\right) \otimes \boldsymbol{X}_n^\top\right)\right)^\top \tag{38}$$

$$\boldsymbol{Z}' = \left(vec\left(\left(\boldsymbol{X}_1^\top \boldsymbol{X}_1\right) \otimes \boldsymbol{X}_1\right), \ldots, vec\left(\left(\boldsymbol{X}_n^\top \boldsymbol{X}_n\right) \otimes \boldsymbol{X}_n\right)\right)^\top \tag{39}$$

*If $\boldsymbol{Z}$ satisfies the 2-RIP condition with constant $\delta_2$, $\boldsymbol{Z}'$ also satisfies the 2-RIP condition with constant $\delta_2$.*

**Proof.**

$$\left\|\boldsymbol{Z}'\mathrm{vec}(\boldsymbol{M})\right\|_2^2$$

$$= \{\mathrm{vec}(\boldsymbol{M})\}^\top \boldsymbol{Z}'^\top \boldsymbol{Z}' \mathrm{vec}(\boldsymbol{M})$$

$$= \sum_{i=1}^n \{\mathrm{vec}(\boldsymbol{M})\}^\top \mathrm{vec}\left(\left(\boldsymbol{X}_i^\top \boldsymbol{X}_i\right) \otimes \boldsymbol{X}_i\right) \left\{\mathrm{vec}\left(\left(\boldsymbol{X}_i^\top \boldsymbol{X}_i\right) \otimes \boldsymbol{X}_i\right)\right\}^\top \mathrm{vec}(\boldsymbol{M})$$

$$= \sum_{i=1}^n \left\langle \boldsymbol{M}, \left(\boldsymbol{X}_i^\top \boldsymbol{X}_i\right) \otimes \boldsymbol{X}_i \right\rangle^2$$

$$= \sum_{i=1}^n \left\langle \boldsymbol{M}^\top, \left(\boldsymbol{X}_i^\top \boldsymbol{X}_i\right) \otimes \boldsymbol{X}_i^\top \right\rangle^2$$

$$= \sum_{i=1}^n \left\{\mathrm{vec}\left(\boldsymbol{M}^\top\right)\right\}^\top \mathrm{vec}\left(\left(\boldsymbol{X}_i^\top \boldsymbol{X}_i\right) \otimes \boldsymbol{X}_i^\top\right) \left\{\mathrm{vec}\left(\left(\boldsymbol{X}_i^\top \boldsymbol{X}_i\right) \otimes \boldsymbol{X}_i^\top\right)\right\}^\top \mathrm{vec}\left(\boldsymbol{M}^\top\right)$$

$$= \left\|\boldsymbol{Z}\mathrm{vec}\left(\boldsymbol{M}^\top\right)\right\|_2^2$$

According to RIP condition on $\boldsymbol{Z}$,

$$(1-\delta_2)\|\boldsymbol{M}\|_F^2 = (1-\delta_2)\left\|\boldsymbol{M}^\top\right\|_F^2 \leq \frac{1}{n}\left\|\boldsymbol{Z}\mathrm{vec}\left(\boldsymbol{M}^\top\right)\right\|_2^2 \leq (1+\delta_2)\left\|\boldsymbol{M}^\top\right\|_F^2 = (1+\delta_2)\|\boldsymbol{M}\|_F^2$$

It follows that

$$(1-\delta_2)\|\boldsymbol{M}\|_F^2 \leq \frac{1}{n}\left\|\boldsymbol{Z}'\mathrm{vec}(\boldsymbol{M})\right\|_2^2 \leq (1+\delta_2)\|\boldsymbol{M}\|_F^2$$

which indicates that $\boldsymbol{Z}'$ satisfies the same 2-RIP condition as $\boldsymbol{Z}$. $\qquad\square$

**Lemma C.4.** *Suppose $\boldsymbol{Z}$ satisfies the 2-RIP condition with constant $\delta_2$. For two matrices $\boldsymbol{M}_1$ and $\boldsymbol{M}_2$, we have*

$$|\langle\boldsymbol{Z}vec(\boldsymbol{M}_1), \boldsymbol{Z}vec(\boldsymbol{M}_2)\rangle - \langle\boldsymbol{M}_1, \boldsymbol{M}_2\rangle| \leq 3\delta_2\|\boldsymbol{M}_1\|_F\|\boldsymbol{M}_2\|_F \tag{40}$$

**Proof.** Due to RIP condition, we directly have $\|\boldsymbol{Z}\mathrm{vec}(\boldsymbol{M}_1 + \boldsymbol{M}_2)\|_2^2 \leq (1+\delta_2)\|\boldsymbol{M}_1 + \boldsymbol{M}_2\|_F^2$, which can be expanded as

$$\|\boldsymbol{Z}\mathrm{vec}(\boldsymbol{M}_1)\|_F^2 + \|\boldsymbol{Z}\mathrm{vec}(\boldsymbol{M}_2)\|_F^2 + 2\langle\boldsymbol{Z}\mathrm{vec}(\boldsymbol{M}_1), \boldsymbol{Z}\mathrm{vec}(\boldsymbol{M}_2)\rangle$$
$$\leq (1+\delta_2)(\|\boldsymbol{M}_1\|_F^2 + \|\boldsymbol{M}_2\|_F^2 + 2\langle\boldsymbol{M}_1, \boldsymbol{M}_2\rangle)$$

Again due to RIP condition, we also have

$$(1-\delta_2)\|\boldsymbol{M}_1\|_F^2 \leq \|\boldsymbol{Z}\mathrm{vec}(\boldsymbol{M}_1)\|_2^2 \quad\text{and}\quad (1-\delta_2)\|\boldsymbol{M}_2\|_F^2 \leq \|\boldsymbol{Z}\mathrm{vec}(\boldsymbol{M}_2)\|_2^2$$

Consequently, it holds that

$$(1-\delta_2)(\|\boldsymbol{M}_1\|_F^2 + \|\boldsymbol{M}_1\|_F^2) + 2\langle\boldsymbol{Z}\mathrm{vec}(\boldsymbol{M}_1), \boldsymbol{Z}\mathrm{vec}(\boldsymbol{M}_2)\rangle$$
$$\leq (1+\delta_2)(\|\boldsymbol{M}_1\|_F^2 + \|\boldsymbol{M}_2\|_F^2 + 2\langle\boldsymbol{M}_1, \boldsymbol{M}_2\rangle)$$

Namely,

$$\langle\boldsymbol{Z}\mathrm{vec}(\boldsymbol{M}_1), \boldsymbol{Z}\mathrm{vec}(\boldsymbol{M}_2)\rangle - \langle\boldsymbol{M}_1, \boldsymbol{M}_2\rangle \leq \delta_2(\|\boldsymbol{M}_1\|_F^2 + \|\boldsymbol{M}_2\|_F^2 + \langle\boldsymbol{M}_1, \boldsymbol{M}_2\rangle)$$

Furthermore, we note that the last inequality still holds if we replace $\boldsymbol{M}_1$ by $\lambda\boldsymbol{M}_1$ and $\boldsymbol{M}_2$ by $1/\lambda\boldsymbol{M}_2$. Optimizing the RHS with $\lambda$, we get

$$\langle\boldsymbol{Z}\mathrm{vec}(\boldsymbol{M}_1), \boldsymbol{Z}\mathrm{vec}(\boldsymbol{M}_2)\rangle - \langle\boldsymbol{M}_1, \boldsymbol{M}_2\rangle \leq 3\delta_2\|\boldsymbol{M}_1\|_F\|\boldsymbol{M}_2\|_F$$

Proving the other side of the inequality is similar. $\qquad\square$

**Lemma C.5.** *Let $\boldsymbol{Z}_i = \left(\boldsymbol{X}_i^\top \boldsymbol{X}_i\right) \otimes \boldsymbol{X}_i^\top$. With $\|\tilde{\boldsymbol{d}}\|_2 = \|\boldsymbol{d}^*\|_2 = 1$, denote $\check{\boldsymbol{\Sigma}}$ and $\hat{\boldsymbol{\Sigma}}$ respectively as*

$$\check{\boldsymbol{\Sigma}} = \sum_{i=1}^n \boldsymbol{Z}_i\tilde{\boldsymbol{d}}\tilde{\boldsymbol{d}}^\top \boldsymbol{Z}_i^\top, \quad \hat{\boldsymbol{\Sigma}} = \sum_{i=1}^n \boldsymbol{Z}_i\tilde{\boldsymbol{d}}\left(\boldsymbol{d}^*\right)^\top \boldsymbol{Z}_i^\top.$$

*Then we have*

$$\left\|\check{\boldsymbol{\Sigma}}^{-1}\left(\langle\tilde{\boldsymbol{d}}, \boldsymbol{d}^*\rangle\check{\boldsymbol{\Sigma}} - \hat{\boldsymbol{\Sigma}}\right)\right\|_2 \leq \frac{3\delta_2}{1-3\delta_2}dist\left(\tilde{\boldsymbol{d}}, \boldsymbol{d}^*\right) \tag{41}$$

**Proof.** First consider the minimal eigenvalue of $\check{\boldsymbol{\Sigma}}$

$$
\begin{aligned}
\lambda_{\min}\left(\check{\boldsymbol{\Sigma}}\right) &= \min_{\|\boldsymbol{u}\|_2=1} \boldsymbol{u}^\top \check{\boldsymbol{\Sigma}} \boldsymbol{u} \\
&= \min_{\|\boldsymbol{u}\|_2=1} \sum_{i=1}^n \boldsymbol{u}^\top \boldsymbol{Z}_i \tilde{\boldsymbol{d}} \tilde{\boldsymbol{d}}^\top \boldsymbol{Z}_i^\top \boldsymbol{u} \\
&= \min_{\|\boldsymbol{u}\|_2=1} \sum_{i=1}^n \text{tr}\left(\boldsymbol{u}^\top \boldsymbol{Z}_i \tilde{\boldsymbol{d}}\right) \text{tr}\left(\boldsymbol{u}^\top \boldsymbol{Z}_i \tilde{\boldsymbol{d}}\right) \\
&= \min_{\|\boldsymbol{u}\|_2=1} \sum_{i=1}^n \left(\left\langle \boldsymbol{Z}_i, \boldsymbol{u}\tilde{\boldsymbol{d}}^\top \right\rangle\right)^2 \\
&= \min_{\|\boldsymbol{u}\|_2=1} \left\|\boldsymbol{Z}\text{vec}\left(\boldsymbol{u}\tilde{\boldsymbol{d}}^\top\right)\right\|_2^2 \\
&\geq 1 - 3\delta_2
\end{aligned}
$$

The inequality holds due to Lemma C.4.

Further consider

$$
\begin{aligned}
\left\|\langle\tilde{\boldsymbol{d}},\boldsymbol{d}^*\rangle\check{\boldsymbol{\Sigma}} - \hat{\boldsymbol{\Sigma}}\right\|_2 &= \max_{\|\boldsymbol{u}\|_2=\|\boldsymbol{v}\|_2=1} \boldsymbol{u}^\top \left(\langle\tilde{\boldsymbol{d}},\boldsymbol{d}^*\rangle\check{\boldsymbol{\Sigma}} - \hat{\boldsymbol{\Sigma}}\right)\boldsymbol{v} \\
&= \max_{\|\boldsymbol{u}\|_2=\|\boldsymbol{v}\|_2=1} \sum_{i=1}^n \left(\langle\tilde{\boldsymbol{d}},\boldsymbol{d}^*\rangle\boldsymbol{u}^\top \boldsymbol{Z}_i \tilde{\boldsymbol{d}}\tilde{\boldsymbol{d}}^\top \boldsymbol{Z}_i^\top \boldsymbol{v} - \boldsymbol{u}^\top \boldsymbol{Z}_i \tilde{\boldsymbol{d}}\left(\boldsymbol{d}^*\right)^\top \boldsymbol{Z}_i^\top \boldsymbol{v}\right) \\
&= \max_{\|\boldsymbol{u}\|_2=\|\boldsymbol{v}\|_2=1} \sum_{i=1}^n \left\langle \boldsymbol{Z}_i, \boldsymbol{u}\tilde{\boldsymbol{d}}^\top\right\rangle \left\langle \boldsymbol{Z}_i, \langle\tilde{\boldsymbol{d}},\boldsymbol{d}^*\rangle\boldsymbol{v}\tilde{\boldsymbol{d}}^\top\right\rangle - \left\langle \boldsymbol{Z}_i, \boldsymbol{u}\tilde{\boldsymbol{d}}^\top\right\rangle \left\langle \boldsymbol{Z}_i, \boldsymbol{v}\left(\boldsymbol{d}^*\right)^\top\right\rangle \\
&= \max_{\|\boldsymbol{u}\|_2=\|\boldsymbol{v}\|_2=1} \sum_{i=1}^n \left\langle \boldsymbol{Z}_i, \boldsymbol{u}\tilde{\boldsymbol{d}}^\top\right\rangle \left\langle \boldsymbol{Z}_i, \boldsymbol{v}\left(\langle\tilde{\boldsymbol{d}},\boldsymbol{d}^*\rangle\tilde{\boldsymbol{d}} - \boldsymbol{d}^*\right)^\top\right\rangle \\
&= \left\langle \boldsymbol{Z}\text{vec}\left(\boldsymbol{u}\tilde{\boldsymbol{d}}^\top\right), \boldsymbol{Z}\text{vec}\left(\boldsymbol{v}\left(\langle\tilde{\boldsymbol{d}},\boldsymbol{d}^*\rangle\tilde{\boldsymbol{d}} - \boldsymbol{d}^*\right)^\top\right)\right\rangle \\
&\leq 3\delta_2 \left\|\langle\tilde{\boldsymbol{d}},\boldsymbol{d}^*\rangle\tilde{\boldsymbol{d}} - \boldsymbol{d}^*\right\|_2 + \left\langle \boldsymbol{u}\tilde{\boldsymbol{d}}^\top, \boldsymbol{v}\left(\langle\tilde{\boldsymbol{d}},\boldsymbol{d}^*\rangle\tilde{\boldsymbol{d}} - \boldsymbol{d}^*\right)^\top\right\rangle \\
&= 3\delta_2\text{dist}(\tilde{\boldsymbol{d}},\boldsymbol{d}^*)
\end{aligned}
$$

The inequality holds due to Lemma C.4 where the inner product equals to 0 because $\tilde{\boldsymbol{d}} \perp \langle\tilde{\boldsymbol{d}},\boldsymbol{d}^*\rangle\tilde{\boldsymbol{d}} - \boldsymbol{d}^*$. $\qquad\square$

## C.2 Proof of Theorem 5.2

For ease of display, we present a prerequisite theorem before proof of Theorem 5.2. The following theorem provides error bounds within each iteration, which is the key to prove Theorem 5.2.

**Theorem C.6.** *Suppose model (21) holds and solved by alternating minimization algorithm. Assume Condition 5.1 with a small constant $\delta_2$. Let $\kappa_1 = \mu_0 + 3\delta_2/(1-3\delta_2)$ and $\kappa_2 = \mu_0 + (3\delta_2+\lambda_2)/(1-3\delta_2+\lambda_2)$. Then we have*

$$
dist\left(\hat{\boldsymbol{w}}^{(t)}, \boldsymbol{w}\right) \leq \kappa_1 dist\left(\hat{\boldsymbol{d}}^{(t)}, \boldsymbol{d}\right) + \tau_1 \tag{42}
$$

$$
dist\left(\hat{\boldsymbol{d}}^{(t)}, \boldsymbol{d}\right) \leq \kappa_2 dist\left(\hat{\boldsymbol{w}}^{(t-1)}, \boldsymbol{w}\right) + \tau_2 \tag{43}
$$

**Proof.** The procedures of proofs for (42) and (43) are the same, with some differences in details.

Let us focus on (42) first. Then the model (12) can be rewritten in matrix form as below

$$
\boldsymbol{Y} = \boldsymbol{M}\boldsymbol{w} + \varepsilon
$$

with $\boldsymbol{w} = \mathrm{vec}(\boldsymbol{W})$ and $\boldsymbol{M}$ defined as follows

$$\boldsymbol{M} = \left( \mathrm{vec}\left( \boldsymbol{X}_1^\top \boldsymbol{D} \boldsymbol{X}_1^\top \boldsymbol{X}_1 \right), \dots, \mathrm{vec}\left( \boldsymbol{X}_n^\top \boldsymbol{D} \boldsymbol{X}_n^\top \boldsymbol{X}_n \right) \right)^\top \tag{44}$$

Suppose $\boldsymbol{w}$ and $\widehat{\boldsymbol{w}}^{(t)}$ are normalized, so we have $\|\boldsymbol{w}\|_2 = \|\widehat{\boldsymbol{w}}^{(t)}\|_2 = 1$ for any $t \geq 1$ in the following. Let $\tilde{\boldsymbol{d}}^{(t)} = \widehat{\boldsymbol{d}}^{(t)}/\|\widehat{\boldsymbol{d}}^{(t)}\|_2$ and $\boldsymbol{d}^* = \boldsymbol{d}/\|\boldsymbol{d}\|_2$ be their unit vectors. Define two matrices in $t$-th iterations

$$\check{\boldsymbol{\Sigma}}^{(t)} = \sum_{i=1}^n \boldsymbol{Z}_i \tilde{\boldsymbol{d}}^{(t)} \left( \tilde{\boldsymbol{d}}^{(t)} \right)^\top \boldsymbol{Z}_i^\top, \quad \hat{\boldsymbol{\Sigma}}^{(t)} = \sum_{i=1}^n \boldsymbol{Z}_i \tilde{\boldsymbol{d}}^{(t)} (\boldsymbol{d}^*)^\top \boldsymbol{Z}_i^\top.$$

Define $\widehat{\boldsymbol{M}}^{(t)}$ as (44) with $\boldsymbol{D}$ replaced by $\widehat{\boldsymbol{D}}^{(t)}$. Then, it holds that

$$\left( \widehat{\boldsymbol{M}}^{(t)} \right)^\top \widehat{\boldsymbol{M}}^{(t)} = \|\widehat{\boldsymbol{d}}^{(t)}\|_2^2 \check{\boldsymbol{\Sigma}}^{(t)} \text{ and } \left( \widehat{\boldsymbol{M}}^{(t)} \right)^\top \boldsymbol{M} = \|\widehat{\boldsymbol{d}}^{(t)}\|_2 \|\boldsymbol{d}\|_2 \hat{\boldsymbol{\Sigma}}^{(t)}$$

Given $\widehat{\boldsymbol{D}}^{(t)}$, we have

$$\begin{aligned}
\widehat{\boldsymbol{w}}^{(t)} &= \left\{ \left( \widehat{\boldsymbol{M}}^{(t)} \right)^\top \widehat{\boldsymbol{M}}^{(t)} \right\}^{-1} \left( \widehat{\boldsymbol{M}}^{(t)} \right)^\top \boldsymbol{Y} \\
&= \left\{ \left( \widehat{\boldsymbol{M}}^{(t)} \right)^\top \widehat{\boldsymbol{M}}^{(t)} \right\}^{-1} \left( \widehat{\boldsymbol{M}}^{(t)} \right)^\top (\boldsymbol{M}\boldsymbol{w} + \boldsymbol{\varepsilon}) \\
&= \frac{\|\boldsymbol{d}\|_2}{\|\widehat{\boldsymbol{d}}^{(t)}\|_2} \left( \check{\boldsymbol{\Sigma}}^{(t)} \right)^{-1} \hat{\boldsymbol{\Sigma}}^{(t)} + \frac{1}{\|\widehat{\boldsymbol{d}}^{(t)}\|_2^2} \left( \check{\boldsymbol{\Sigma}}^{(t)} \right)^{-1} \left( \widehat{\boldsymbol{M}}^{(t)} \right)^\top \boldsymbol{\varepsilon}
\end{aligned}$$

Without loss of generality, suppose $\langle \widehat{\boldsymbol{d}}^{(t)}, \boldsymbol{d} \rangle \geq 0$. The case that $\langle \widehat{\boldsymbol{d}}^{(t)}, \boldsymbol{d} \rangle < 0$ can be proved in a similar way. Consider the $\ell_2$-norm distance

$$\begin{aligned}
& \left\| \frac{\|\widehat{\boldsymbol{d}}^{(t)}\|_2}{\|\boldsymbol{d}\|_2} \widehat{\boldsymbol{w}}^{(t)} - \boldsymbol{w} \right\|_2 \\
&= \left( \check{\boldsymbol{\Sigma}}^{(t)} \right)^{-1} \hat{\boldsymbol{\Sigma}}^{(t)} \boldsymbol{w} - \boldsymbol{w} + \frac{1}{\|\widehat{\boldsymbol{d}}^{(t)}\|_2 \|\boldsymbol{d}\|_2} \left( \check{\boldsymbol{\Sigma}}^{(t)} \right)^{-1} \left( \widehat{\boldsymbol{M}}^{(t)} \right)^\top \boldsymbol{\varepsilon} \\
&= \langle \tilde{\boldsymbol{d}}^{(t)}, \boldsymbol{d}^* \rangle \boldsymbol{w} - \boldsymbol{w} - \left( \check{\boldsymbol{\Sigma}}^{(t)} \right)^{-1} \left( \langle \tilde{\boldsymbol{d}}^{(t)}, \boldsymbol{d}^* \rangle \check{\boldsymbol{\Sigma}}^{(t)} - \hat{\boldsymbol{\Sigma}}^{(t)} \right) \boldsymbol{w} + \frac{1}{\|\widehat{\boldsymbol{d}}^{(t)}\|_2 \|\boldsymbol{d}\|_2} \left( \check{\boldsymbol{\Sigma}}^{(t)} \right)^{-1} \left( \widehat{\boldsymbol{M}}^{(t)} \right)^\top \boldsymbol{\varepsilon} \\
&\leq \underbrace{1 - \langle \tilde{\boldsymbol{d}}^{(t)}, \boldsymbol{d}^* \rangle}_{A1} + \underbrace{\left\| \left( \check{\boldsymbol{\Sigma}}^{(t)} \right)^{-1} \left( \langle \tilde{\boldsymbol{d}}^{(t)}, \boldsymbol{d}^* \rangle \check{\boldsymbol{\Sigma}}^{(t)} - \hat{\boldsymbol{\Sigma}}^{(t)} \right) \right\|_2}_{A2} + \underbrace{\left\| \frac{1}{\|\widehat{\boldsymbol{d}}^{(t)}\|_2 \|\boldsymbol{d}\|_2} \left( \check{\boldsymbol{\Sigma}}^{(t)} \right)^{-1} \left( \widehat{\boldsymbol{M}}^{(t)} \right)^\top \boldsymbol{\varepsilon} \right\|_2}_{A3}
\end{aligned} \tag{45}$$

Note that when $\langle \widehat{\boldsymbol{d}}^{(t)}, \boldsymbol{d} \rangle \geq 0$, we have $0 \leq \langle \tilde{\boldsymbol{d}}^{(t)}, \boldsymbol{d}^* \rangle \leq 1$. Thus for A1,

$$1 - \langle \tilde{\boldsymbol{d}}^{(t)}, \boldsymbol{d}^* \rangle \leq 1 - \langle \tilde{\boldsymbol{d}}^{(t)}, \boldsymbol{d}^* \rangle^2 = \mathrm{dist}^2\left( \widehat{\boldsymbol{d}}^{(t)}, \boldsymbol{d} \right) \leq \mu_0 \mathrm{dist}\left( \widehat{\boldsymbol{d}}^{(t)}, \boldsymbol{d} \right)$$

For A2, it holds that according to Lemma C.5

$$A2 \leq \frac{3\delta_2}{1 - 3\delta_2} \mathrm{dist}\left( {}^{(t)}, \boldsymbol{d} \right)$$

For A3, first note that $\|\check{\boldsymbol{\Sigma}}^{(t)}\|_2 \geq 1 - 3\delta_2$.

$$\left\| \left( \widehat{\boldsymbol{M}}^{(t)} \right)^\top \boldsymbol{\varepsilon} \right\|_2 = \left\| \sum_{i=1}^n \varepsilon_i \mathrm{vec} \left( \boldsymbol{X}_i^\top \widehat{\boldsymbol{D}}^{(t)} \boldsymbol{X}_i^\top \boldsymbol{X}_i \right) \right\|_2$$

$$\leq \sup \left\{ \left\| \sum_{i=1}^n \varepsilon_i \mathrm{vec} \left( \boldsymbol{X}_i^\top \widehat{\boldsymbol{D}}^{(t)} \boldsymbol{X}_i^\top \boldsymbol{X}_i \right) \right\|_2 \right\}$$

$$= \tau_0$$

As a result, A3 can be bounded by

$$\mathrm{A3} \leq \frac{\tau_0}{(1 - 3\delta_2)\|\widehat{\boldsymbol{d}}^{(t)}\|_2 \|\boldsymbol{d}\|_2} = \tau_1$$

One the other hand, according to Lemma C.1 we have for any $c > 0$ that

$$\mathrm{dist}\left( \widehat{\boldsymbol{w}}^{(t)}, \boldsymbol{w} \right) = \mathrm{dist}\left( c\widehat{\boldsymbol{w}}^{(t)}, \boldsymbol{w} \right) \leq \left\| c\widehat{\boldsymbol{w}}^{(t)} - \boldsymbol{w} \right\|_2$$

Therefore,

$$\mathrm{dist}\left( \widehat{\boldsymbol{w}}^{(t)}, \boldsymbol{w} \right) \leq \left( \mu_0 + \frac{3\delta_2}{1 - 3\delta_2} \right) \mathrm{dist}\left( \widehat{\boldsymbol{d}}^{(t)}, \boldsymbol{d} \right) + \tau_1 \tag{46}$$

To prove (43), first we need to rewrite the model (12) in another matrix form below.

$$\boldsymbol{Y} = \boldsymbol{N}\boldsymbol{d} + \boldsymbol{\varepsilon}$$

with $\boldsymbol{d} = \mathrm{vec}(\boldsymbol{D})$ and $\boldsymbol{N}$ defined as follows

$$\boldsymbol{N} = \left( \mathrm{vec}\left( \boldsymbol{X}_1 \boldsymbol{W} \boldsymbol{X}_1^\top \boldsymbol{X}_1 \right), \ldots, \mathrm{vec}\left( \boldsymbol{X}_n \boldsymbol{W} \boldsymbol{X}_n^\top \boldsymbol{X}_n \right) \right)^\top \tag{47}$$

Note that $\|\boldsymbol{w}\|_2 = \|\widehat{\boldsymbol{w}}^{(t-1)}\|_2 = 1$. Define two matrices in $t$-th iterations

$$\check{\boldsymbol{\Sigma}}^{(t-1)} = \sum_{i=1}^n \boldsymbol{Z}_i' \widehat{\boldsymbol{w}}^{(t-1)} \left( \widehat{\boldsymbol{w}}^{(t-1)} \right)^\top \left( \boldsymbol{Z}_i' \right)^\top, \quad \widehat{\boldsymbol{\Sigma}}^{(t-1)} = \sum_{i=1}^n \boldsymbol{Z}_i' \widehat{\boldsymbol{w}}^{(t-1)} \boldsymbol{w}^\top \left( \boldsymbol{Z}_i' \right)^\top.$$

Define $\widehat{\boldsymbol{N}}^{(t-1)}$ as (47) with $\boldsymbol{W}$ replaced by $\widehat{\boldsymbol{W}}^{(t-1)}$. Then, it holds that

$$\left( \widehat{\boldsymbol{N}}^{(t-1)} \right)^\top \widehat{\boldsymbol{N}}^{(t-1)} = \check{\boldsymbol{\Sigma}}^{(t-1)} \text{ and } \left( \widehat{\boldsymbol{N}}^{(t-1)} \right)^\top \boldsymbol{N} = \widehat{\boldsymbol{\Sigma}}^{(t-1)}$$

Given $\widehat{\boldsymbol{W}}^{(t-1)}$, we have

$$\widehat{\boldsymbol{d}}^{(t)} = \left\{ \left( \widehat{\boldsymbol{N}}^{(t-1)} \right)^\top \widehat{\boldsymbol{N}}^{(t-1)} + \lambda_2 \boldsymbol{I} \right\}^{-1} \left( \widehat{\boldsymbol{N}}^{(t-1)} \right)^\top (\boldsymbol{N}\boldsymbol{d} + \boldsymbol{\varepsilon})$$

$$= \left( \check{\boldsymbol{\Sigma}}^{(t-1)} + \lambda_2 \boldsymbol{I} \right)^{-1} \widehat{\boldsymbol{\Sigma}}^{(t-1)} \boldsymbol{d} + \left( \check{\boldsymbol{\Sigma}}^{(t-1)} + \lambda_2 \boldsymbol{I} \right)^{-1} \left( \widehat{\boldsymbol{N}}^{(t-1)} \right)^\top \boldsymbol{\varepsilon}$$

$$= \langle \widehat{\boldsymbol{w}}^{(t-1)}, \boldsymbol{w} \rangle \boldsymbol{d} - \left( \check{\boldsymbol{\Sigma}}^{(t-1)} + \lambda_2 \boldsymbol{I} \right)^{-1} \left( \langle \widehat{\boldsymbol{w}}^{(t-1)}, \boldsymbol{w} \rangle \left( \check{\boldsymbol{\Sigma}}^{(t-1)} + \lambda_2 \boldsymbol{I} \right) - \widehat{\boldsymbol{\Sigma}}^{(t-1)} \right) \boldsymbol{d}$$

$$+ \left( \check{\boldsymbol{\Sigma}}^{(t-1)} + \lambda_2 \boldsymbol{I} \right)^{-1} \left( \widehat{\boldsymbol{N}}^{(t-1)} \right)^\top \boldsymbol{\varepsilon}$$

Then $\ell_2$-norm error of $\widehat{\boldsymbol{d}}^{(t)}$ can be also bounded in a similar way. Take the case $\langle \widehat{\boldsymbol{w}}^{(t-1)}, \boldsymbol{w} \rangle \geq 0$ for example.

$$
\begin{aligned}
&\frac{\|\widehat{\boldsymbol{d}}^{(t)} - \boldsymbol{d}\|_2}{\|\boldsymbol{d}\|_2} \\
&\leq \underbrace{1 - \langle \widehat{\boldsymbol{w}}^{(t-1)}, \boldsymbol{w} \rangle}_{B1} + \underbrace{\left\| \left( \check{\boldsymbol{\Sigma}}^{(t-1)} + \lambda_2 \boldsymbol{I} \right)^{-1} \left( \langle \boldsymbol{w}^{(t-1)}, \boldsymbol{w} \rangle \left( \check{\boldsymbol{\Sigma}}^{(t-1)} + \lambda_2 \boldsymbol{I} \right) - \widehat{\boldsymbol{\Sigma}}^{(t-1)} \right) \right\|_2}_{B2} \\
&\quad + \underbrace{\frac{1}{\|\boldsymbol{d}\|_2} \left\| \left( \check{\boldsymbol{\Sigma}}^{(t-1)} + \lambda_2 \boldsymbol{I} \right)^{-1} \left( \widehat{\boldsymbol{N}}^{(t-1)} \right)^{\top} \boldsymbol{\varepsilon} \right\|_2}_{B3}
\end{aligned}
\tag{48}
$$

Resembling A1, A2 and A3, we have

$$
B1 = 1 - \langle \widehat{\boldsymbol{w}}^{(t-1)}, \boldsymbol{w} \rangle \leq \mu_0 \mathrm{dist} \left( \widehat{\boldsymbol{w}}^{(t-1)}, \widehat{\boldsymbol{w}} \right)
\tag{49}
$$

$$
B2 \leq \frac{3\delta_2 + \lambda_2}{1 - 3\delta_2 + \lambda_2} \mathrm{dist} \left( \widehat{\boldsymbol{w}}^{(t-1)}, \widehat{\boldsymbol{w}} \right)
\tag{50}
$$

$$
B3 \leq \frac{1}{1 - 3\delta_2 + \lambda_2} \|\boldsymbol{\varepsilon}\|_2 = \tau_2
\tag{51}
$$

Thus we have

$$
\mathrm{dist} \left( \widehat{\boldsymbol{d}}^{(t)}, \boldsymbol{d} \right) \leq \left( \mu_0 + \frac{3\delta_2 + \lambda_2}{1 - 3\delta_2 + \lambda_2} \right) \mathrm{dist} \left( \widehat{\boldsymbol{w}}^{(t-1)}, \boldsymbol{w} \right) + \tau_2
\tag{52}
$$

Last we need to prove by induction that if $\mathrm{dist} \left( \widehat{\boldsymbol{w}}^{(0)}, \boldsymbol{w} \right) \leq \mu_0$ and $\mu_0$ satisfies $\kappa_1 \mu_0 + \tau_1 \leq \mu_0$ and $\kappa_2 \mu_0 + \tau_2 \leq \mu_0$, then $\mathrm{dist} \left( \widehat{\boldsymbol{w}}^{(t)}, \boldsymbol{w} \right) \leq \mu_0$ and $\mathrm{dist} \left( \widehat{\boldsymbol{d}}^{(t)}, \boldsymbol{d} \right) \leq \mu_0$ for any $t \geq 1$.

When $t = 1$,

$$
\begin{aligned}
\mathrm{dist} \left( \widehat{\boldsymbol{d}}^{(1)}, \boldsymbol{d} \right) &\leq \mathrm{dist}^2 \left( \widehat{\boldsymbol{w}}^{(0)}, \boldsymbol{w} \right) + \frac{3\delta_2 + \lambda_2}{1 - 3\delta_2 + \lambda_2} \mathrm{dist} \left( \widehat{\boldsymbol{w}}^{(0)}, \boldsymbol{w} \right) + \tau_2 \\
&\leq \left( \mu_0 + \frac{3\delta_2 + \lambda_2}{1 - 3\delta_2 + \lambda_2} \right) \mathrm{dist} \left( \widehat{\boldsymbol{w}}^{(0)}, \boldsymbol{w} \right) + \tau_2 \\
&\leq \kappa_2 \mu_0 + \tau_2 \\
&\leq \mu_0
\end{aligned}
$$

Furthermore,

$$
\begin{aligned}
\mathrm{dist} \left( \widehat{\boldsymbol{w}}^{(1)}, \boldsymbol{w} \right) &\leq \mathrm{dist}^2 \left( \widehat{\boldsymbol{d}}^{(1)}, \boldsymbol{d} \right) + \frac{3\delta_2}{1 - 3\delta_2} \mathrm{dist} \left( \widehat{\boldsymbol{d}}^{(1)}, \boldsymbol{d} \right) + \tau_1 \\
&\leq \left( \mu_0 + \frac{3\delta_2}{1 - 3\delta_2} \right) \mathrm{dist} \left( \widehat{\boldsymbol{d}}^{(1)}, \boldsymbol{d} \right) + \tau_1 \\
&\leq \kappa_1 \mu_0 + \tau_1 \\
&\leq \mu_0
\end{aligned}
$$

This completes the proof of initial step $t = 1$.

When $t \geq 2$, suppose dist $\left(\widehat{\boldsymbol{w}}^{(t-1)}, \boldsymbol{w}\right) \leq \mu_0$ to prove the $t$-th case.

$$\begin{aligned} \text{dist}\left(\widehat{\boldsymbol{d}}^{(t)}, \boldsymbol{d}\right) &\leq \text{dist}^2\left(\widehat{\boldsymbol{w}}^{(t-1)}, \boldsymbol{w}\right) + \frac{3\delta_2 + \lambda_2}{1 - 3\delta_2 + \lambda_2}\text{dist}\left(\widehat{\boldsymbol{w}}^{(t-1)}, \boldsymbol{w}\right) + \tau_2 \\ &\leq \left(\mu_0 + \frac{3\delta_2 + \lambda_2}{1 - 3\delta_2 + \lambda_2}\right)\text{dist}\left(\widehat{\boldsymbol{w}}^{(t-1)}, \boldsymbol{w}\right) + \tau_2 \\ &\leq \kappa_2\mu_0 + \tau_2 \\ &\leq \mu_0 \end{aligned}$$

Furthermore,

$$\begin{aligned} \text{dist}\left(\widehat{\boldsymbol{w}}^{(t)}, \boldsymbol{w}\right) &\leq \text{dist}^2\left(\widehat{\boldsymbol{d}}^{(t)}, \boldsymbol{d}\right) + \frac{3\delta_2}{1 - 3\delta_2}\text{dist}\left(\widehat{\boldsymbol{d}}^{(t)}, \boldsymbol{d}\right) + \tau_1 \\ &\leq \left(\mu_0 + \frac{3\delta_2}{1 - 3\delta_2}\right)\text{dist}\left(\widehat{\boldsymbol{d}}^{(t)}, \boldsymbol{d}\right) + \tau_1 \\ &\leq \kappa_1\mu_0 + \tau_1 \\ &\leq \mu_0 \end{aligned}$$

This completes the induction. In words, the distances dist $\left(\widehat{\boldsymbol{w}}^{(t)}, \boldsymbol{w}\right)$ and dist $\left(\widehat{\boldsymbol{d}}^{(t)}, \boldsymbol{d}\right)$ in all iterations are guaranteed to not exceed the initial distance $\mu_0$, which is required when proving (42) and (43). The proof is now complete. □

**Proof of Theorem 5.2**

Theorem C.6 provides the error bounds within an iteration. Therefore, we have the following by recursion,

$$\begin{aligned} \text{dist}\left(\widehat{\boldsymbol{w}}^{(t)}, \boldsymbol{w}\right) &\leq \kappa_1\text{dist}\left(\widehat{\boldsymbol{d}}^{(t)}, \boldsymbol{d}\right) + \tau_1 \\ &\leq (\kappa_1\kappa_2)\text{dist}\left(\widehat{\boldsymbol{w}}^{(t-1)}, \boldsymbol{w}\right) + \kappa_1\tau_2 + \tau_1 \\ &\leq (\kappa_1\kappa_2)^t\text{dist}\left(\widehat{\boldsymbol{w}}^{(0)}, \boldsymbol{w}\right) + \sum_{s=0}^{t-1}(\kappa_1\kappa_2)^s(\kappa_1\tau_2 + \tau_1) \\ &\leq (\kappa_1\kappa_2)^t\mu_0 + \frac{\kappa_1\tau_2 + \tau_1}{1 - \kappa_1\kappa_2} \end{aligned}$$

On the other hand,

$$\begin{aligned} \text{dist}\left(\widehat{\boldsymbol{d}}^{(t)}, \boldsymbol{d}\right) &\leq \kappa_2\text{dist}\left(\widehat{\boldsymbol{w}}^{(t-1)}, \boldsymbol{w}\right) + \tau_2 \\ &\leq (\kappa_1\kappa_2)\text{dist}\left(\widehat{\boldsymbol{d}}^{(t-1)}, \boldsymbol{d}\right) + \kappa_2\tau_1 + \tau_2 \\ &\leq (\kappa_1\kappa_2)^{t-1}\text{dist}\left(\widehat{\boldsymbol{d}}^{(1)}, \boldsymbol{d}\right) + \sum_{s=0}^{t-2}(\kappa_1\kappa_2)^s(\kappa_2\tau_1 + \tau_2) \\ &\leq \kappa_1^{t-1}\kappa_2^t\mu_0 + \frac{\kappa_2\tau_1 + \tau_2}{1 - \kappa_1\kappa_2} \end{aligned}$$

The proof is completed. □

## C.3 Proof of Theorem 5.3

**Theorem C.7.** *Suppose model (21) holds and solved by alternating minimization algorithm. Assume Condition 5.1 with a small constant $\delta_2$. Denote $c^{(t)} = \|\widehat{\boldsymbol{d}}^{(t)}\|_2/\|\boldsymbol{d}\|_2$. Let $\nu_1 = 2\mu_0 + 3\delta_2/(1 - 3\delta_2)$*

*and $\nu_2 = 2\mu_0 + (3\delta_2 + \lambda_2)/(1 - 3\delta_2 + \lambda_2)$. Then for any $t \geq 0$ we have*

$$\left\| c^{(t)} \widehat{\boldsymbol{w}}^{(t)} - \boldsymbol{w} \right\|_2 \leq (\nu_1 \nu_2)^t \mu_0 + \frac{\nu_1 \tau_2 + \tau_1}{1 - \nu_1 \nu_2} \tag{53}$$

$$\frac{\|\widehat{\boldsymbol{d}}^{(t)} - \boldsymbol{d}\|_2}{\|\boldsymbol{d}\|_2} \leq \nu_1^{t-1} \nu_2^t \mu_0 + \frac{\nu_2 \tau_1 + \tau_2}{1 - \nu_1 \nu_2} \tag{54}$$

**Proof.** The proof of Theorem C.7 is also completed by induction, resembling that of Theorem 5.2. Thus we just note some key inequalities that are different in induction procedures. Suppose for any $t \geq 1$ that $\|\widehat{\boldsymbol{d}}^{(t)} - \boldsymbol{d}\|_2 / \|\boldsymbol{d}\|_2 \leq \mu_0$ and $\|c^{(t)} \widehat{\boldsymbol{w}}^{(t)} - \boldsymbol{w}\|_2 \leq \mu_0$.

According to (48) we have

$$
\begin{aligned}
\frac{\|\widehat{\boldsymbol{d}}^{(t)} - \boldsymbol{d}\|_2}{\|\boldsymbol{d}\|_2} &\leq \frac{1}{2} \|\widehat{\boldsymbol{w}}^{(t-1)} - \boldsymbol{w}\|_2^2 + \frac{3\delta_2 + \lambda_2}{1 - 3\delta_2 + \lambda_2} \mathrm{dist}\left(\widehat{\boldsymbol{w}}^{(t-1)}, \boldsymbol{w}\right) + \tau_2 \\
&\leq 2\|c^{(t-1)} \widehat{\boldsymbol{w}}^{(t-1)} - \boldsymbol{w}\|_2^2 + \frac{3\delta_2 + \lambda_2}{1 - 3\delta_2 + \lambda_2} \|c^{(t-1)} \widehat{\boldsymbol{w}}^{(t-1)} - \boldsymbol{w}\|_2 + \tau_2 \\
&\leq \left(2\mu_0 + \frac{3\delta_2 + \lambda_2}{1 - 3\delta_2 + \lambda_2}\right) \|c^{(t-1)} \widehat{\boldsymbol{w}}^{(t-1)} - \boldsymbol{w}\|_2 + \tau_2
\end{aligned}
\tag{55}
$$

The second inequality holds because Lemma C.2.

On the other hand, according to (45) we have

$$
\begin{aligned}
\left\| c^{(t)} \widehat{\boldsymbol{w}}^{(t)} - \boldsymbol{w} \right\|_2 &\leq \frac{1}{2} \|\tilde{\boldsymbol{d}}^{(t)} - \boldsymbol{d}^*\|_2^2 + \frac{3\delta_2}{1 - 3\delta_2} \mathrm{dist}\left(\tilde{\boldsymbol{d}}^{(t)}, \boldsymbol{d}^*\right) + \tau_1 \\
&\leq 2\frac{\|\widehat{\boldsymbol{d}}^{(t)} - \boldsymbol{d}\|_2^2}{\|\boldsymbol{d}\|_2^2} + \frac{3\delta_2}{1 - 3\delta_2} \frac{\|\widehat{\boldsymbol{d}}^{(t)} - \boldsymbol{d}\|_2}{\|\boldsymbol{d}\|_2} + \tau_1 \\
&\leq \left(2\mu_0 + \frac{3\delta_2}{1 - 3\delta_2}\right) \frac{\|\widehat{\boldsymbol{d}}^{(t)} - \boldsymbol{d}\|_2}{\|\boldsymbol{d}\|_2} + \tau_1
\end{aligned}
\tag{56}
$$

Given (55) and (56), $\ell_2$-norm errors can be also bounded in $t$-th iteration. The remaining proof can be completed in the same way as Theorem 5.2. $\square$

**Proof of Theorem 5.3**

According to Condition 5.1, firstly we have

$$\|\widehat{\boldsymbol{Y}}^{(t)} - \boldsymbol{Y}\|_2 = \left\| \boldsymbol{Z} \mathrm{vec}\left( \widehat{\boldsymbol{w}}^{(t)} \left(\widehat{\boldsymbol{d}}^{(t)}\right)^\top - \boldsymbol{w} \boldsymbol{d}^\top \right) \right\|_2 \leq \sqrt{1 + \delta_2} \left\| \widehat{\boldsymbol{w}}^{(t)} \left(\widehat{\boldsymbol{d}}^{(t)}\right)^\top - \boldsymbol{w} \boldsymbol{d}^\top \right\|_F$$

It follows that

$$\left\| \widehat{\boldsymbol{w}}^{(t)} \left( \widehat{\boldsymbol{d}}^{(t)} \right)^{\top} - \boldsymbol{w} \boldsymbol{d}^{\top} \right\|_F$$

$$= \left\| \widehat{\boldsymbol{w}}^{(t)} \left( \widehat{\boldsymbol{d}}^{(t)} - \boldsymbol{d} \right)^{\top} + \left( \widehat{\boldsymbol{w}}^{(t)} - \boldsymbol{w} \right) \boldsymbol{d}^{\top} \right\|_F$$

$$\leq \| \widehat{\boldsymbol{d}}^{(t)} - \boldsymbol{d} \|_2 + \| \boldsymbol{d} \|_2 \left\| \widehat{\boldsymbol{w}}^{(t)} - \boldsymbol{w} \right\|_2$$

$$\leq \| \boldsymbol{d} \|_2 \frac{\| \widehat{\boldsymbol{d}}^{(t)} - \boldsymbol{d} \|_2}{\| \boldsymbol{d} \|_2} + \| \boldsymbol{d} \|_2 \frac{2}{c^{(t)} + 1} \left\| c^{(t)} \widehat{\boldsymbol{w}}^{(t)} - \boldsymbol{w} \right\|_2$$

$$\leq \| \boldsymbol{d} \|_2 \left( \frac{\| \widehat{\boldsymbol{d}}^{(t)} - \boldsymbol{d} \|_2}{\| \boldsymbol{d} \|_2} + 2 \left\| c^{(t)} \widehat{\boldsymbol{w}}^{(t)} - \boldsymbol{w} \right\|_2 \right)$$

$$\leq \| \boldsymbol{d} \|_2 \left\{ \left( (\nu_1 \nu_2)^{t-1} \mu_0 + \frac{\nu_1 \tau_2 + \tau_1}{1 - \nu_1 \nu_2} \right) + 2 \left( (\nu_1 \nu_2)^{t-1} \mu_0 + \frac{\nu_2 \tau_1 + \tau_2}{1 - \nu_1 \nu_2} \right) \right\}$$

$$\leq 3 \| \boldsymbol{d} \|_2 \left( (\nu_1 \nu_2)^{t-1} \mu_0 + \frac{\tau_1 + \tau_2}{1 - \nu_1 \nu_2} \right)$$

Therefore we have

$$\| \widehat{\boldsymbol{Y}}^{(t)} - \boldsymbol{Y} \|_2 \leq 3 \| \boldsymbol{D} \|_F \sqrt{1 + \delta_2} \left\{ (\nu_1 \nu_2)^{t-1} \mu_0 + \frac{\tau_1 + \tau_2}{1 - \nu_1 \nu_2} \right\}$$

$\square$

