# OpenReview forum: "Attention boosted Individualized Regression"
_NeurIPS.cc/2024/Conference — NeurIPS 2024 poster_

### Official Review · Reviewer_gg3f · 2024-06-26

**Soundness:** 3
**Presentation:** 4
**Contribution:** 4
**Rating:** 8
**Confidence:** 4

**Summary:**

The authors propose an individualised matrix regression method, where the individualised part is shown to be related to self-attention. The method is presented nicely, with theoretical and empirical results that indicate the usefulness of the method.

**Strengths:**

The paper is well-written and easy to follow. The method is clear, the connection to attention is clear, the theoretical results appear correct and the empirical results (on both simulated and real data) and convincing.

**Weaknesses:**

- The introduction mentions personalised interpretations, but that is never illustrated in the paper (though seems to be alluded to in the appendix?)
 - Proposition 1 seems to assume that the function g is an element-wise function (or otherwise a particular function, such that it can be transposed), but it is mentioned as being any general function.
 - The subscripts K and Q appear to have been swapped in the definition (I) of W. See line 176 also.
 - The convergence is geometric, but while one term disappears, there appears to still be a constant term left. So the question would be how large that constant is? How tight is the achieved bound in the limit as t -> \infty?
 - It would be better to report mean and _standard error_ (standard deviation of the mean) instead of mean and standard deviation of the 100 repetitions. This makes it easier to compare the results between methods.
 - Equation 18 is presented without any constraints on W, but the pseudo-code normalises the weight matrix (Equation 31), and the proofs assume unit vectors. Are these the same? If not, dot he convergence proofs still hold? This should be clarified.
 - I'm missing a discussion on the computational complexity and run-time (and compared to other methods).

**Questions:**

- When directly referring to references, use the Firstauthorlastname et al.~\cite{ref} format. Should be possible to use \citet{ref} for this.
 - Make equations part of sentences instead of something particular presented following a colon. Also, when equations are at the end of a sentence, end it with a full stop.
 - Equation 1: The d_1,d_2 subscript is missing from R.
 - Line 161: The functio \rho is not explained properly/clearly. Is this a notational error?
 - It appears you are using the Frobenius matrix inner product, but it is never defined. Same with the vector inner product, seems like you are using the Euclidean inner product (dot product), but this is never defined.
 - Line 206: What do you mean by: "... has the potential to achieve fewer parameters and faster training."?
 - In the theorems: Define W in the initial distance.
 - Explain what you mean by "the truth" on line 228.
 - Some spelling errors and typos, e.g. lines 239, 269, 498, and 535.
 - References: RNNs should be upper case in Ref 12, "Lasso" in Ref 25, and reference 24 has some problems with the spacing (copy from PDF?).
 - Label the first axis in Figure 3.
 - Lines 424-437: Should be "Python", "Matlab", and "Pytorch", since they are names.

**Limitations:**

- See my previous comment about personalised interpretation.
 - The discussion section is _very_ short. I understand the problem with space constratins, but do elaborate properly on strengths, weaknesses, limitations, and future work.
 - You say that Crowdsourcing data from human subjects and IRB approval is not applicable, but these are images from human subjects collected collaboratively in the ADNI project. You would actually likely need an IRB approval to perform this research, since you do research on data from humans, but it seems that doing research on public medical data is a gray area.

---

> ### Author Rebuttal · Authors · 2024-08-07
>
> Thank you very much for your review.
>
> # Main questions
>
> >**Q1**: Personalised interpretations
>
> **A1**: Thanks for your feedback. For real data, we show in Figure 2 the individualized coefficients of different samples and their significant internal relations and frame out corresponding regions in original samples. Take sample 1 for example. The block (4, 5) has the strongest relations, and is related to both (6, 4) and (6, 5), indicating the important relations between corpus callosum and hippocampus. We also find that after separating heterogeneous effects, the homogeneous effects highlight the hippocampus region, which is widely known to be associated with Alzheimer's disease. We will add the discussion in the revised version.
>
> >**Q2**: The function $g$ in Proposition 1.
>
> **A2**: Thank you for question. Sorry for the confusion in Proposition 1. Yes, we are mostly focusing on element-wise $g$ function, including commonly used activation functions in attention mechanisms, such as row-wise softmax function, scaling function and more. We will clarify it in the revised version.
>
>
> >**Q3**: The convergence is geometric, but while one term disappears, there appears to still be a constant term left. So the question would be how large that constant is? How tight is the achieved bound in the limit as t -> \infty?
>
> **A3**: Many thanks for the good question. When the error term is sub-Gaussian, it can be proved that after $t\ge t_0 + \log(\log(n)/n)/(2\log(\kappa))$ iterations, the distance errors can be bounded by $\mathcal{O}(\sqrt{\log(n) / n})$ with high probability, where $t_0$ is a constant and $\kappa$ is the contraction parameter. We will remark this specifically in the revised version.
>
> >**Q4**: Equation 18 is presented without any constraints on W, but the pseudo-code normalises the weight matrix (Equation 31), and the proofs assume unit vectors. Are these the same? If not, do the convergence proofs still hold? This should be clarified.
>
> **A4**: Many thanks for your question. We apologize for the oversight. The Equation 18 should also require a norm constraint for W due to identifiability consideration. Therefore, this is consistent with the proof (where an unit vector assumption is imposed). We will update Equation 18 in the revisions.
>
> >**Q5**: Computational complexity and run-time.
>
> **A5**: Thanks for your feedback. As the algorithm actually alternatively solves linear models, the complexity is $O(D_1^3D_2^3)$ where $D_1$, $D_2$ are the size of images. On the other  hand, the optimization problem is non-convex, but it is bi-convex. Theorem 5.2 suggests a linear convergence rate of the alternating minimization algorithm (AMA) although the problem is non-convex. In contrast, gradient descent (GD) algorithms could achieve a linear convergence rate only if the objective function is strongly convex. But for general non-convex functions, linear convergence cannot be achieved. This suggests the advantage of AMA over GD. We will add the discussion on the computational complexity and report the run-time compared to other methods in the revised version. Thanks again for your advice.
>
> >**Q6**: Line 161: The function \rho is not explained properly/clearly. Is this a notational error?
>
> **A6**: Thanks for your question. Here $\rho$ stands for the linear function considered in the reference. Specifically, for a matrix input $\boldsymbol{Y}\in \mathbb{R}^{n\times p}$, define $\rho(\cdot)$ as $\rho(\boldsymbol{Y}) = \boldsymbol{Y} / n$. We will make it clear when revising.
>
> >**Q7**: Line 206: What do you mean by: "... has the potential to achieve fewer parameters and faster training."?
>
> **A7**: This sentence is to introduce the above reference titled ``Simplifying transformer blocks’’, in which the proposed simplified transformers enjoy faster training speed using fewer parameters. This further suggests the advantage of simplified models.
>
> >**Q8**: Explain what you mean by "the truth" on line 228.
>
> **A8**: The truth means true counterparts of $\boldsymbol{W}^{(t)}$ and $\boldsymbol{D}^{(t)}$, i.e. $\boldsymbol{W}$ and $\boldsymbol{D}$ in the true model.
>
> >**Q9**: References: RNNs should be upper case in Ref 12, "Lasso" in Ref 25, and reference 24 has some problems with the spacing (copy from PDF?).
>
> **A9**: The information on references were downloaded from Google Scholar directly, where there may be some issues. Thank you for pointing them out and we will make adjustments in the update.
>
> ## Minor revisions
> * The subscripts K and Q appear to have been swapped in the definition (I) of W. See line 176 also.
> * It would be better to report mean and standard error (standard deviation of the mean) instead of mean and standard deviation of the 100 repetitions.
> * When directly referring to references, use the Firstauthorlastname et al.~\cite{ref} format. Should be possible to use \citet{ref} for this.
> * Make equations part of sentences instead of something particular presented following a colon. Also, when equations are at the end of a sentence, end it with a full stop.
> * Equation 1: The d_1,d_2 subscript is missing from R.
> * Define Frobenius inner product and Euclidean inner product used.
> * In the theorems: Define W in the initial distance.
> * Some spelling errors and typos, e.g. lines 239, 269, 498, and 535.
> * Label the first axis in Figure 3.
> * Lines 424-437: Should be "Python", "Matlab", and "Pytorch", since they are names.
> * Crowdsourcing data from human subjects and IRB approval.
>
> Many thanks to your carefulness. We will make adjustments according to the above suggestions and update in the revised version. Thank you again.

---

> > ### Comment · Reviewer_gg3f · 2024-08-09
> > **Thank you for the updates and explanations**
> >
> > I appreciate the authors' efforts to explain and improve the paper. The rebuttal addresses most of my concerns.

---

### Official Review · Reviewer_hyyZ · 2024-07-12

**Soundness:** 3
**Presentation:** 3
**Contribution:** 3
**Rating:** 6
**Confidence:** 2

**Summary:**

The paper introduces a method for self-attention-based individualized regression and derives it's relation to transformers. The method is evaluated in a simulation setting and on an Alzheimer Brain MRI dataset.

**Strengths:**

- Interesting theoretical treatment of individualized regression and its connection to transformers
- Well written

**Weaknesses:**

- Only applied to tiny datasets with image size 48 x 48
- Only two experiments

**Questions:**

- It would be great if the authors could comment on the scalability of the method
- Is individualized regression applicable to multiple instance learning problems?

**Limitations:**

See questions

---

> ### Author Rebuttal · Authors · 2024-08-07
>
> Thank you very much for your review.
>
> >**Q1**: Only applied to tiny datasets with image size 48 x 48 and only two experiments.
>
> **A1**: Thank you for your feedback. The MRI scans are preprocessed to be of size $113\times 137\times 113$ and we further resize the extracted slices to the size $48\times 48$ for computational efficiency.
> Besides, we have added a 5-fold cross-validation in real study to test the significance of the difference, which shows that the advantage of the proposed method is significant, with results shown in the following table.
>
> | Methods | AIR | LRMR | TRLasso | DKN | ViT |
> | :--------: | :--------: | :--------: | :--------: | :--------: | :--------: |
> | Fold 1 | 3.183 | 3.712 | 3.359 | 3.297 | 3.321 |
> | Fold 2 | 3.193 | 3.744 | 3.325 | 3.283 | 3.349 |
> | Fold 3 | 3.127 | 3.699 | 3.258 | 3.284 | 3.215 |
> | Fold 4 | 3.129 | 3.700 | 3.232 | 3.226 | 3.282 |
> | Fold 5 | 3.093 | 3.717 | 3.286 | 3.214 | 3.243 |
> | Mean (S.D.) | 3.145 (0.042) | 3.715 (0.018) | 3.292 (0.051) | 3.261 (0.038) | 3.282 (0.055) |
> | P-value | - | 0.000 | 0.001 | 0.002 | 0.002 |
>
> In the coming days, we will attempt to apply our method to new real data to further validate its performance.
>
> >**Q2**: It would be great if the authors could comment on the scalability of the method.
>
> **A2**: Thanks for your suggestion. Two aspects of scalability are usually concerned.
> * Computational efficiency: As the algorithm actually alternatively solves linear models, the complexity is $O(D_1^3D_2^3)$ where $D_1$, $D_2$ are the size of images. As for the convergence rate, if we further suppose noise $\epsilon_i$ is sub-gaussian, it can be proved that after $t\ge t_0 + \log(\log(n)/n)/(2\log(\kappa))$ iterations, the distance errors can be bounded by $\mathcal{O}(\sqrt{\log(n) / n})$ with high probability, where $t_0$ is a constant and $\kappa$ is the contraction parameter. In summary, as the size of the dataset increases, larger images size brings more computational burden, which can be relieved by parallel processing and more in practice. On the other hand, larger sample size leads to fewer iterations and smaller errors.
> * Generalization: First we propose to combine the homogeneous and heterogeneous parts to make our model adaptive to more types of data. Second, we propose to model the internal relation matrix by a function $g$, which can introduce nonlinearity to make the model more flexible. However, as mentioned in the paper, the ability of the model to handle general data is limited, depending on the gap between the model and real cases, which is a common issue of model-based methods.
> Thanks for your suggestion again and we will add the discussion in the revised version.
>
> >**Q3**: Is individualized regression applicable to multiple instance learning problems?
>
> **A3**: Thanks for the interesting question. In multiple instance learning problems, the training data is organized into bags, where each bag contains multiple instances. In the context of MIL, individualized regression can be applied by considering each bag as an individual data point and tailoring regression models to these bags. Our model can be also applied to multiple instance learning problems, in the sense that patches of an image are instances in a bag and internal relations determine bag-specific coefficients.

---

> > ### Comment · Reviewer_hyyZ · 2024-08-10
> >
> > Thank you for your response!

---

### Official Review · Reviewer_EsRE · 2024-07-15

**Soundness:** 3
**Presentation:** 3
**Contribution:** 3
**Rating:** 6
**Confidence:** 4

**Summary:**

This paper proposed an individualized regression method and applied it to medical image analysis. The method can handle matrix-valued data and does not require additional information on sample similarity. The authors also analyzed its relationship to the attention technique. Finally, the proposed method was evaluated on simulation and real data sets, and obtained improved performance.

**Strengths:**

1. A novel individualized regression method handling one-model-fit-all issue.
2. The method does not require additional information on sample similarity.

**Weaknesses:**

1. The method can only work for matrix-valued data, such as image data.
2. The real study is too simple and insufficient.
3. Section 2 should be rewritten to make it clearer since some technique details of the proposed method are hard to understand.

**Questions:**

1. Section 2 should be rewritten to make it clearer since some technique details of the proposed method are hard to understand.
2. Too many details of the method are unclear, such as the inverse operation of R, p1 *p2, and d1 * d2.
3. How to determine the size of the blocks? How to determine the number of factors?
4. Please double check Eq. (14).

**Limitations:**

The authors did not state the limitations of the proposed method. No conclusion of this paper.

---

> ### Author Rebuttal · Authors · 2024-08-07
>
> Thank you very much for your review.
>
> >**Q1**: The method can only work for matrix-valued data, such as image data.
>
> **A1**: Thanks for your comment. Matrix-valued data, particularly images, are pervasive in many practical applications, and our method aims to provide a novel solution for these scenarios. While as the amount of tensor data increases, extending the method to tensor-valued data is a valuable direction to expand its applications. We will consider how to generalize our model in future research, including how to deal with higher dimensions, how to incorporate internal relations into the individualized coefficients, and more.
>
> >**Q2**: The real study is too simple and insufficient.
>
> **A2**: Thank you for your feedback. We have added a 5-fold cross-validation in real study to test the significance of the difference. It shows that the advantage of the proposed method is significant, of which results are in the following table.
>
> | Methods | AIR | LRMR | TRLasso | DKN | ViT |
> | :--------: | :--------: | :--------: | :--------: | :--------: | :--------: |
> | Fold 1 | 3.183 | 3.712 | 3.359 | 3.297 | 3.321 |
> | Fold 2 | 3.193 | 3.744 | 3.325 | 3.283 | 3.349 |
> | Fold 3 | 3.127 | 3.699 | 3.258 | 3.284 | 3.215 |
> | Fold 4 | 3.129 | 3.700 | 3.232 | 3.226 | 3.282 |
> | Fold 5 | 3.093 | 3.717 | 3.286 | 3.214 | 3.243 |
> | Mean (S.D.) | 3.145 (0.042) | 3.715 (0.018) | 3.292 (0.051) | 3.261 (0.038) | 3.282 (0.055) |
> | P-value | - | 0.000 | 0.001 | 0.002 | 0.002 |
>
> In the coming days, we will attempt to apply our method to new real data to further validate its performance.
>
> >**Q3**: Section 2 should be rewritten to make it clearer since some technique details of the proposed method are hard to understand.
>
> **A3**: Thank you for your feedback. We will revise Section 2 to enhance clarity, particularly providing more technique details of the proposed method to make it clearer.
>
> >**Q4**: Too many details of the method are unclear, such as the inverse operation of R, p1 *p2, and d1 * d2.
>
> **A4**: Thanks for pointing it out. The inverse operation of $\mathcal{R}$ is used to recover the reshaped images and corresponding coefficients to their original reshape. Besides, $p_1\times p_2$ are the number of blocks and $d_1\times d_2$ are the size of blocks. We have scrutinized the details and will make them clearer in the revised version.
>
> >**Q5**: How to determine the size of the blocks? How to determine the number of factors?
>
> **A5**: Thanks for your question. The division of images before implementation of our model is similar to that of Vision Transformer, where $16\times 16$ is a common size of patches. In our paper, due to relevantly small size of images, the size of blocks should be smaller. We note that our method performs robust to moderately small size of blocks, so we determine the size of blocks by cross-validation among $4\times 4$, $6\times 6$ and $8\times 8$.
> Besides, when the row-wise internal relations are of interest, such as EEG in which each row represents a channel, it will not involve division and the method can be directly applied.
>
> >**Q6**: Please double check Eq. (14).
>
> **A6**: Thanks for pointing it out. There is a typo that the first $W$ should be transposed. It is tantamount that we assume $W^T$ in the model, which has no effect on the existing results. Thank you again.
>
> > **Q7**: The authors did not state the limitations of the proposed method. No conclusion of this paper.
>
> **A7**: Thanks for your comments. We conclude in the “Discussion” section and will add a paragraph about limitations there, stating that:
> “On the other hand, we realize that the AIR framework also has limitations. First, the AIR model is designed for data with heterogeneous internal relationships, and its capability to handle more general data is more or less restricted. When there are minimal heterogeneous effects, its performance will be similar to an ordinary linear model. Second, as discussed earlier, our framework could be viewed as a simplified version of the Vision Transformer; however, such simplifications may also reduce its approximation power for more complex scenarios. Furthermore, this paper primarily investigates the linear form of AIR. Although the linear form performs well in the cases of interest, exploring the generalization of the model in future work is still worthwhile.”

---

> > ### Comment · Reviewer_EsRE · 2024-08-13
> >
> > I thank the authors for their responses. However, some concerns remain unsolved such as the number of factors, the new real data, and the size blocks just considered based on data-driven. So, my score remains unchanged.

---

> > > ### Author Response · Authors · 2024-08-13
> > >
> > > Thanks for your response. Given the size of images $D_1, D_2$, the number of factors $p_1, p_2$ are determined as long as the size of blocks $d_1, d_2$ are determined because they need to satisfy $(p_1, p_2) = (D_1/d_2, D_2/d_2)$. Due to the short period of revision, we did not implement new real data but will consider that hereafter. However, the new implementation with cross-validation could demonstrate the robustness of our approach. Thank you again.

---

### Official Review · Reviewer_3WVs · 2024-07-19

**Soundness:** 2
**Presentation:** 3
**Contribution:** 3
**Rating:** 6
**Confidence:** 2

**Summary:**

The paper proposes an approach for regression where common model coefficients can be modulated by sample-specific data. In particular, here the approach is applied to images (or matrices), where sample-specific data is derived from patch similarities (measured through rotation correlation), reflecting intra-image homogeneity. The model parameters (matrices of coefficients) are learned through penalised least-squares, with energy-minimising Frobenius norms on the unknown coefficient matrices to make the problem more well-posed. The authors then draw correspondances between their model and self-attention under some assumptions. They then show analytically convergence rates and error bounds for their model and the AD-style optimisation algorithm. Finally, simulation results and empirical results on brain imaging data show lower prediction errors compared to related methods.

**Strengths:**

The paper provides an interesting connection between varying-coefficient models and self-attention under relatively mild conditions.

The ablation studies in appendix B.1 is interesting and helps highlight the contribution of individual coefficients in different cases.

The method proposed has applications for images and other matrix-valued data.

The figures are very helpful in conveying the aspects of the coefficient matrices.

**Weaknesses:**

Claims of superiority are not supported by hypothesis tests between the proposed method and the other 4 methods. (post-rebuttal: this is now OK)

For real data analysis, it is unclear how many subjects were selected, and with which diagnosis. Extracting 10 slice per subject is fine, but are they evaluated as independent or are results provided per-subject? (post-rebuttal: OK)

In addition, predicting cognitive scores from brain imaging is a very well studied task (in particular the MMSE-ADNI combination), not only cross-sectionally but also longitudinally. See e.g. 10.1016/j.neuroimage.2011.09.069, 10.1016/j.neuroimage.2014.03.036, 10.1109/PRNI.2015.28, or the review in 10.1016/j.jalz.2016.11.007. Here, the choice of the metric 'improvement compared to sd of scores in test set' obscures performance with respect to these and other previous work. I would suggest to provide MSE and MAE in the original MMSE scale for more clarity. (post-rebuttal: OK)

**Questions:**

Is the performance of the method proposed significantly different from the other methods, both for the simulation and the real data case? (post-rebuttal: OK)

How is the lack of independence between slices from the same subject addressed in testing, in particular for cross-validation and error metric computation? (post-rebuttal: OK)

**Limitations:**

Limitations are not in a separate section and consist of one sentence. (post-rebuttal: OK)

There is no real negative social impact of this work.

---

> ### Author Rebuttal · Authors · 2024-08-07
>
> Thank you very much for your review.
>
> >**Q1**: Is the performance of the method proposed significantly different from the other methods, both for the simulation and the real data case? (Claims of superiority are not supported by hypothesis tests between the proposed method and the other 4 methods.)
>
> **A1**: Thanks for your question. For the simulation where the results are from 100 repetitions, we use z-test showing that the superiority of the proposed method to the other methods is significant.
> As for real brain imaging analysis, we did not conduct significance tests in our original experiments because of the chronological division considered. But to have a rough understanding of the robustness of the proposed method, we retest the performances of these methods by 5-fold cross-validation. The result of 5 folds in real study is shown below.
>
> | Methods | AIR | LRMR | TRLasso | DKN | ViT |
> | :--------: | :--------: | :--------: | :--------: | :--------: | :--------: |
> | Fold 1 | 3.183 | 3.712 | 3.359 | 3.297 | 3.321 |
> | Fold 2 | 3.193 | 3.744 | 3.325 | 3.283 | 3.349 |
> | Fold 3 | 3.127 | 3.699 | 3.258 | 3.284 | 3.215 |
> | Fold 4 | 3.129 | 3.700 | 3.232 | 3.226 | 3.282 |
> | Fold 5 | 3.093 | 3.717 | 3.286 | 3.214 | 3.243 |
> | Mean (S.D.) | 3.145 (0.042) | 3.715 (0.018) | 3.292 (0.051) | 3.261 (0.038) | 3.282 (0.055) |
>
> Based on the mean and S.D. of the obtained RMSE, we may conduct a (rather rough) t-test and report the obtained p-value as below.
>
> | Tests | AIR vs LRMR | AIR vs TRLasso | AIR vs DKN | AIR vs ViT |
> | :--------: | :--------: | :--------: | :--------: | :--------: |
> | P-value | 0.000 | 0.001 | 0.002 | 0.002 |
>
> These results suggest that the proposed method is significantly better than the others.
> We will add the results of significance tests in the revised version.
>
> >**Q2**: The choice of the metric 'improvement compared to sd of scores in test set' obscures performance with respect to these and other previous work. I would suggest to provide MSE and MAE in the original MMSE scale for more clarity.
>
> **A2**: Thanks for your suggestion. We have reorganized the table with RMSE and add the results of hypothesis tests of 5-fold cross-validation for a reference of significance. Please see the table in A1.
>
> >**Q3**: For real data analysis, it is unclear how many subjects were selected, and with which diagnosis. How is the lack of independence between slices from the same subject addressed in testing, in particular for cross-validation and error metric computation?
>
> **A3**: Thanks for your questions. For the training set, the 7270 images are obtained from 727 subjects in the ADNI&GO phases where 229 are normal, 310 are with MCI (mild cognitive impairment) and 188 are with AD. For the test set, the 3320 images are obtained from 332 subjects in the ADNI2 phase where 140 are normal, 91 are with MCI (mild cognitive impairment) and 101 are with AD. We will explain this in the appendix.
>
> Extracting 10 slices can be viewed as a kind of augmentation of the dataset and we take the obtained images as independent samples, which is a common practice in data augmentation.
> Despite dependency, treating them as independent can be useful for training models, as it effectively increases the diversity and size of the dataset.
> As for the dependency issue when testing, in this revision, we conduct another experiment with 1 middle slice per subject for validation and testing (total of 332 subjects), and 10 slices per subject for training. It shows that the result is very close to the previous one, which also demonstrates the robustness of our method.
>
> >**Q4**: Limitations are not in a separate section and consist of one sentence.
>
> **A4**: Thanks for your suggestion. We will add a paragraph about limitations stating that:
> “On the other hand, we realize that the AIR framework also has limitations. First, the AIR model is designed for data with heterogeneous internal relationships, and its capability to handle more general data is more or less restricted. When there are minimal heterogeneous effects, its performance will be similar to an ordinary linear model. Second, as discussed earlier, our framework could be viewed as a simplified version of the Vision Transformer; however, such simplifications may also reduce its approximation power for more complex scenarios. Furthermore, this paper primarily investigates the linear form of AIR. Although the linear form performs well in the cases of interest, exploring the generalization of the model in future work is still worthwhile.”

---

> > ### Comment · Reviewer_3WVs · 2024-08-12
> >
> > Thank you for the improvements, results are clearer now.
> >
> > I could not find the hypothesis test results in the updated paper, please include (in appendix if needed).
> >
> > Also, if using a t-test, it should be a paired t-test not just a two-sample t-test since the split in folds is the same across methods. Nevertheless I re-ran paired t-tests on the data provided here in table A1 and the claim of superiority seems to hold, so I am upgrading my score.
> >
> > Note that caption of table 2 in paper seems incorrect, these should be improvement in RMSE, not sd.

---

> > > ### Author Response · Authors · 2024-08-12
> > >
> > > Many thanks for your reply. According to the rules of the conference we cannot update the manuscript during this period. We will consider the results of the paired t-tests and add them in the appendix. We will also make the other adjustments aforementioned in the revised version. Thank you again.

---

### Official Review · Reviewer_2raH · 2024-07-20

**Soundness:** 3
**Presentation:** 3
**Contribution:** 3
**Rating:** 5
**Confidence:** 4

**Summary:**

The authors present an interesting approach to individualization of regression for heterogenous data. They first set up an individualized model with additive homogenous and heterogenous components containing matrix-valued coefficient matrices to be learned. They nicely establish the equivalence of their model under mild assumptions with scaled dot-product attention and linear attention approaches and provide a straightforward alternating minimization scheme. They provide both theoretical analysis (two theorems showing geometric decay of optimization error and prediction error under RIP) and experimental results (one synthetic and one real-world application) for their approach.

**Strengths:**

The paper has a number of strengths. It is well written and relevant to current needs in the precision medicine. Although it is not clear to me that it is necessary, the relation with attention is interesting. The author theory holds under realistic assumptions, and they support their findings with a synthetic and a real world example.

**Weaknesses:**

#### Major Weaknesses
I may have missed the anonymized link, but the authors have not provided any code for review, which I find problematic given the applied nature of the paper and the fact that implementation seems straightforward.
.
The individualized coefficients for the ADNI data show are very rough due to blocking in D_i^ori. Is there some way to ameliorate this in future work?

**Questions:**

####Major questions:

Why is there no anonymized link to code?

Why are the individualized coefficients so blocky for ADNI, and what could be done to mitigate this undesirable effect?

**Limitations:**

A more detailed limitations section should be added in the supplement.

---

> ### Author Rebuttal · Authors · 2024-08-07
>
> Thank you very much for your review.
>
> >**Q1**: Why is there no anonymized link to code?
>
> **A1**: Codes were uploaded with submission to the “Supplementary Material” part as a zip file. Due to the request for anonymity, they are not public at the moment. We will provide the public link in the camera ready.
>
> >**Q2**: Why are the individualized coefficients so blocky for ADNI, and what could be done to mitigate this undesirable effect?
>
>  **A2**: We would like to clarify that the blocky effect is essentially caused due to the reshaping operation (from $X_i^{\text{ori}}$ to $X_i$). The reshaping operator $\mathcal{R}(\cdot)$ divides original images into blocks, vectorizes and stacks them for preprocessing.
> The internal relation matrix $A_i$ aggregates blocks in $D^{\text{ori}}$, then we obtain the individualized coefficients $D_i^{\text{ori}}$. This is the reason that $D_i^{\text{ori}}$ appears to be blocky. On the other hand, if we consider internal correlations among small blocks, the blocky effect can be mitigated significantly.
>
> >**Q3**: A more detailed limitations section.
>
> **A3**: Thanks for your suggestion. We will add a paragraph about limitations in the “Discussion” section, stating that:
> “On the other hand, we realize that the AIR framework also has limitations. First, the AIR model is designed for data with heterogeneous internal relationships, and its capability to handle more general data is more or less restricted. When there are minimal heterogeneous effects, its performance will be similar to an ordinary linear model. Second, as discussed earlier, our framework could be viewed as a simplified version of the Vision Transformer; however, such simplifications may also reduce its approximation power for more complex scenarios. Furthermore, this paper primarily investigates the linear form of AIR. Although the linear form performs well in the cases of interest, exploring the generalization of the model in future work is still worthwhile.”

---

> > ### Comment · Reviewer_2raH · 2024-08-12
> > **Thank you.**
> >
> > I thank the reviewers for their responses. My score remains unchanged.

---

### Decision · Program_Chairs · 2024-09-25

**Decision:**

Accept (poster)

**Comment:**

The reviewers appreciated that contribution sheds an interesting theoretical light, relevant for individualized medicine among other things, with fair empirical evaluation, though on somewhat toy data.